# Assembly and seasonality of core phyllosphere microbiota on perennial biofuel crops

Keara L. Grady[1,2,7], Jackson W. Sorensen[1,2,7], Nejc Stopnisek [2,3,7], John Guittar [1,4] & Ashley Shade [1,2,3,5,6]

Perennial grasses are promising feedstocks for biofuel production, with potential for leveraging their native microbiomes to increase their productivity and resilience to environmental stress. Here, we characterize the 16S rRNA gene diversity and seasonal assembly of bacterial and archaeal microbiomes of two perennial cellulosic feedstocks, switchgrass (*Panicum virgatum L.*) and miscanthus (*Miscanthus x giganteus*). We sample leaves and soil every three weeks from pre-emergence through senescence for two consecutive switchgrass growing seasons and one miscanthus season, and identify core leaf taxa based on occupancy. Virtually all leaf taxa are also detected in soil; source-sink modeling shows non-random, ecological filtering by the leaf, suggesting that soil is an important reservoir of phyllosphere diversity. Core leaf taxa include early, mid, and late season groups that were consistent across years and crops. This consistency in leaf microbiome dynamics and core members is promising for microbiome manipulation or management to support crop production.

[1] Department of Microbiology and Molecular Genetics, Michigan State University, 567 Wilson Road, East Lansing, MI 48824, USA. [2] The DOE Great Lakes Bioenergy Research Center, Michigan State University, 1129 Farm Lane, East Lansing, MI 48824, USA. [3] Program in Ecology, Evolutionary Biology and Behavior, Michigan State University, 293 Farm Lane, East Lansing, MI 48824, USA. [4] Kellogg Biological Station, Michigan State University, 3700 E. Gull Lake Dr, Hickory Corners, MI 49060, USA. [5] The Plant Resilience Institute, Michigan State University, East Lansing, MI 48840, USA. [6] Department of Plant, Soil and Microbial Sciences, Michigan State University, East Lansing, MI 48824, USA. [7] These authors Contributed equally: Keara L. Grady, Jackson W. Sorensen, Nejc Stopnisek. Correspondence and requests for materials should be addressed to A.S. (email: shadeash@msu.edu)

The phyllosphere (aerial parts of plants) represents the largest environmental surface area of microbial habitation on the planet[1-3], and much of that surface area is cultivated agriculture, including an estimated $1.5 \times 10^7$ km² of cropland[4]. Phyllosphere microorganisms may provide numerous benefits to plants, including increased stress tolerance[5-7], promotion of growth and reproduction[8-10], protection from foliar pathogens[11], and, with soil microbes, control of flowering phenology[12]. Phyllosphere microorganisms are also thought to play important roles in Earth's biogeochemical cycles by moderating methanol emissions from plants[13,14] and contributing to global nitrogen fixation[15]. Despite this importance, knowledge of phyllosphere microbiomes remains relatively modest, especially for agricultural crops[3,16-18]. To leverage plant microbiomes to support productivity and resilience to environmental stress both above and below ground[19-21], there is a need to advance foundational knowledge of phyllosphere microbiome diversity and dynamics.

Biofuel crops like miscanthus and switchgrass are selected to have extended growing seasons, to produce ample phyllosphere biomass, and to maintain high productivity when grown on marginal lands that are not optimal for food agriculture[22-25]. In the field, these grasses provide extensive leaf habitat, with a seasonal maximum leaf area index (LAI) of 6.2 for switchgrass, and 10 m² leaf surface per m² land for miscanthus[22], as compared to a maximum LAI of 3.2 for corn[26]. Upon senescence, the aboveground biomass is harvested for conversion to biofuels and related bioproducts. Improved understanding of the phyllosphere microbiome is expected to advance goals to predict or manage changes in biomass quality in response to abiotic stress like drought[27-31] or biotic stress like foliar pathogens[32-34].

Leveraging the Great Lakes Bioenergy Research Center's Biofuel Cropping System Experiment (BCSE; a randomized block design established at Michigan State's Kellogg Biological Station in 2008), we ask two questions of the bacterial and archaeal communities (henceforth: microbiomes) inhabiting the leaf surfaces and the associated soils of switchgrass and miscanthus: (1) Are there seasonal patterns of phyllosphere microbiome assembly? If so, are these patterns consistent across fields of the same crop, different crops, and years? (2) To what extent might soil serve as a reservoir of phyllosphere diversity? We find strong seasonal patterns of assembly that is consistent across crops and years, and prioritize a core set of leaf-associated microbiota that are persistent over the season but fluctuate in their relative contributions to the community. Additionally, several lines of evidence suggest that the soil is a key reservoir of leaf microbiota, but also that the leaf habitat selects for particular taxa that generally are not prominent in the soil.

## Results

**Sequencing summary and alpha diversity.** In total, we sequenced 373 phyllosphere epiphyte (leaf surface) and soil samples across the two growing seasons in 2016 and 2017. The number of sequences per sample after our 97% OTU (operational taxonomic unit) clustering pipeline ranged from 20,647 to 359,553. The percentage of sequences belonging to chloroplasts and mitochondria per sample range between 0.2–99.8%, but 235 of the samples (63%) had fewer than 10% chloroplasts and mitochondria reads. After removing sequences that were attributed to chloroplasts and mitochondria or that had unassigned taxonomic classification, we filtered samples that contained fewer than 1000 reads and rarefied the remaining samples to 1000 reads for comparative analyses. While this number of reads is not sufficient to fully capture soil diversity, it does capture phyllosphere diversity (Fig. 1a). The majority of the switchgrass and miscanthus phyllosphere communities were exhaustively

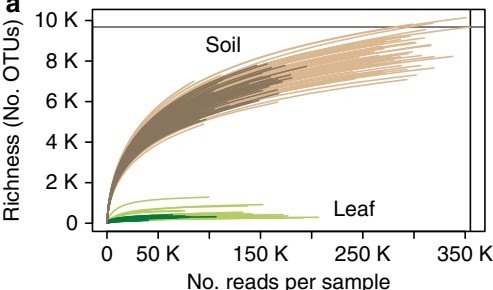

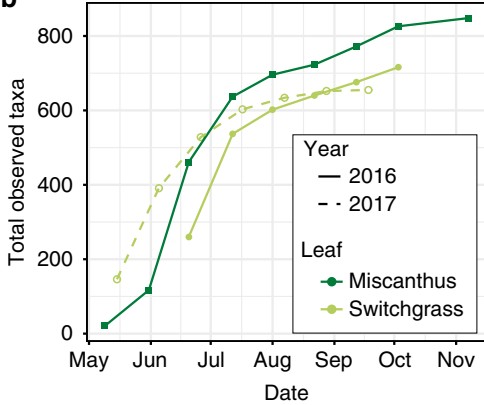

**Fig. 1** Sequencing effort and alpha diversity for switchgrass and miscanthus phyllosphere (green lines) and soils (brown lines). Operational taxonomic units (OTUs) were defined at 97% sequence identity of 16S rRNA gene amplicons. **a** Rarefaction curves of quality-controlled reads. The vertical line is the maximum number of sequences observed in a sample, and the horizontal line is the richness of that sample. **b** Phyllosphere richness accumulation for switchgrass (light green lines and circles) and miscanthus (dark green lines and squares) over time for 2016 (solid lines) and 2017 (dashed lines), using a dataset subsampled to 1000 sequences per sample

sequenced, and approached richness asymptotes with their associated soils.

As reported for other plants[3,35], switchgrass and miscanthus phyllosphere communities had relatively low richness, with 1480 total taxa observed across both crops and consistently fewer than 150 taxa per time point, though there was modest seasonal variability in richness (Supplementary Fig. 1). Cumulative richness increased most between the two earliest time points, and then tapered gradually upward until senescence (Fig. 1b), showing that the contributions of new taxa to community richness were low but consistent over time.

**Seasonal microbiome dynamics.** To perform the most complete temporal analyses of phyllosphere microbiome seasonality, we also subsampled the amplicon sequencing dataset to include the maximum number of time points, resulting in inclusion of 51 discrete leaf and soil samples collected over 18 total time points. The overarching patterns in beta diversity were consistent and statistically indistinguishable from those derived from the same dataset to include more reads per sample but fewer time points (Mantel tests all $p < 0.001$; Supplementary Table 1, Supplementary Table 2, Supplementary Fig. 2). For consistency, we report the patterns from 1000 reads per sample in the main text, but for transparency and comparison, we report results from the minimum reads per sample, inclusive of the complete time series, in supplementary materials.

There were directional seasonal changes in the structures of switchgrass and miscanthus phyllosphere bacterial and archaeal

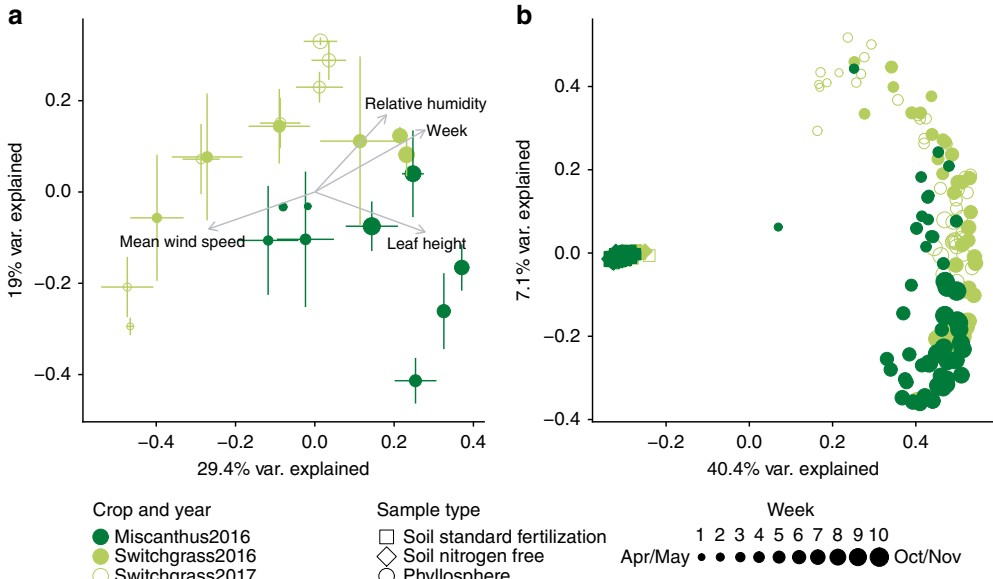

**Fig. 2** Seasonal patterns in the structures of bacterial and archaeal communities inhabiting the phyllosphere and associated soils of the biofuel feedstocks switchgrass and miscanthus. **a** Principal coordinates analysis (PCoA) of switchgrass (light green circles, 2016 = filled and 2017 = open) and miscanthus phyllosphere communities (dark green circles, Bray-Curtis dissimilarity), error bars show 1 deviation around the centroid ($n = 1$ to 8 replicate plots/time point; please see Supplementary Table 1 for exact replicates per time point). **b** PCoA of the phyllosphere communities (circles) relative to the soil (standard fertilization are squares and nitrogen free are diamonds). For both **a** and **b**, subsampling depth was 1000 reads per sample and environmental vectors were fitted when $r^2 > 0.4$ and $p < 0.05$. For both panels, the size of the symbol reflects the sampling week in the season, with smaller symbols used for the earlier time points and larger symbols for the late time points

**Table 1 Permuted multivariate analysis of variance (PERMANOVA) tables for all hypothesis tests for differences in community structure (beta diversity)**

| Dataset | Variable tested | Degrees of freedom | PseudoF | R squared | p-value |
|---|---|---|---|---|---|
| 2016 All | Microbiome habitat (soil v. leaf) | 1 | 166.09 | 0.415 | 0.001 |
| 2017 All | Microbiome habitat (soil v. leaf) | 1 | 74.66 | 0.418 | 0.001 |
| 2016 Phyllosphere | Time | 1 | 21.02 | 0.183 | 0.001 |
| 2016 Soil | Time | 1 | 7.31 | 0.050 | 0.001 |
| 2017 Phyllosphere | Time | 1 | 29.09 | 0.409 | 0.001 |
| 2017 Soil | Time | 1 | 3.05 | 0.048 | 0.001 |
| 2016 Phyllosphere | Crop | 1 | 14.50 | 0.134 | 0.001 |
| 2016 Soil | Crop | 1 | 6.76 | 0.047 | 0.001 |
| 2016 Phyllosphere | Fertilization status | 1 | 0.40 | 0.004 | 0.95 |
| 2016 Soil | Fertilization status | 1 | 4.77 | 0.033 | 0.001 |
| 2017 Phyllosphere | Fertilization status | 1 | 0.87 | 0.020 | 0.438 |
| 2017 Soil | Fertilization status | 1 | 3.41 | 0.054 | 0.001 |

Tests were run individually for each variable

communities (Fig. 2a, Table 1), and these could be attributed to changes in both soil and leaf properties, as well as to weather (Supplementary Table 3). Over the 2016 season, miscanthus and switchgrass phyllosphere communities were synchronous (changed at the same pace and to the same extent, Procrustes m12 = 0.349, $R = 0.807$, $p = 0.021$), and community structure became less variable as the growing season progressed (Supplementary Fig. 3). Switchgrass 2016 and 2017 leaf communities were highly synchronous, suggesting a predictable, interannual assembly (Procrustes m12 = 0.011, $R = 0.994$, $p = 0.008$). The switchgrass community structures were overall equivalent between 2016 and 2017, with the exception of the final time points that were collected post-senescence. Together with the species accumulation analysis (Fig. 1b), these data suggest that these phyllosphere communities are not stochastically assembled, nor are they a linear accumulation over seasonal leaf exposure to whatever taxa are dispersed. The communities follow a directional assembly

over the growing season, and the assembly was highly consistent over 2 years in the switchgrass.

**Contribution of soil microorganisms to phyllosphere assembly.** The major sources of microorganisms to the phyllosphere are soils[2], the vascular tissue of the plant or its seed[36], and the atmosphere or arthropod vectors[3]. As several studies have shown that soil microbes contribute to the phyllosphere microbiome[35,37], we wanted to understand the potential for soil as a reservoir of microorganisms inhabiting switchgrass and miscanthus phyllospheres. We hypothesized that the intersect of shared soil and phyllosphere taxa would be highest early in the season, after the young grasses emerged through the soil. Our deep sequencing effort also provided the opportunity to investigate differences in taxon relative abundances between soil and leaf communities, and to understand what contributions, if any, the soil rare biosphere has for leaf assembly.

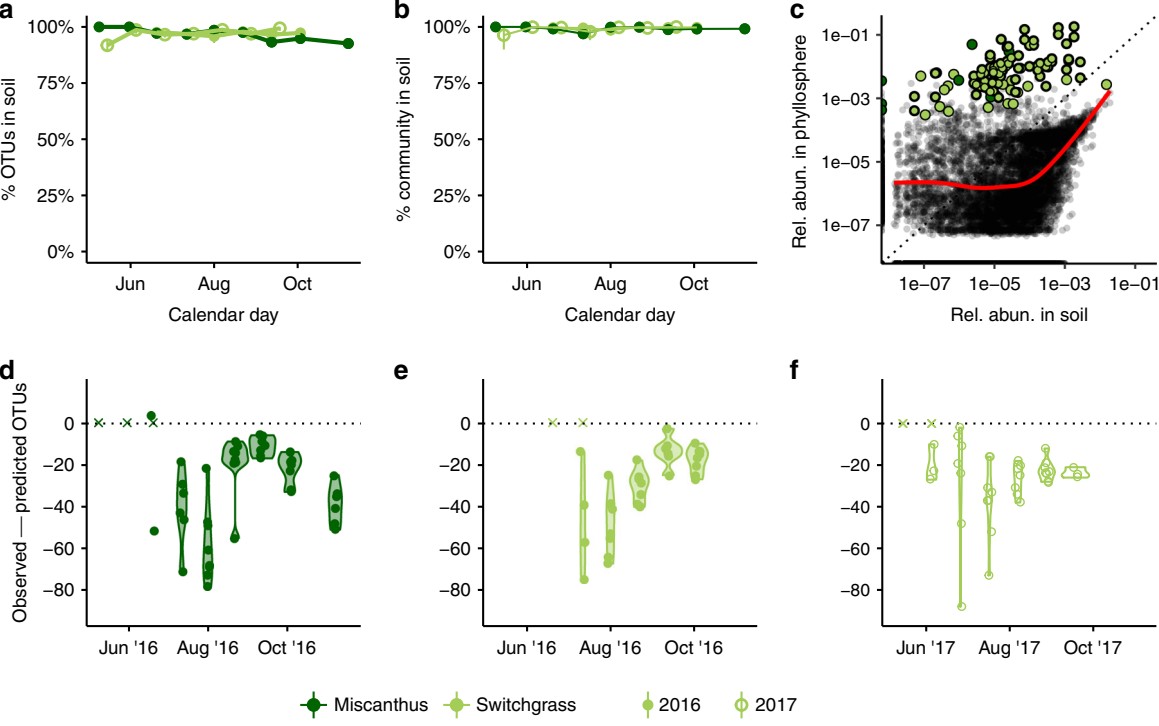

**Fig. 3** The majority of phyllosphere taxa were also present in the soil. **a** Circles represent the mean number of OTUs found in up to eight replicate phyllosphere samples, subsampled to 1000 sequences, for each crop (miscanthus is dark green, filled circles, switchgrass is light green) at each time point (2016 is filled and 2017 is open). An OTU was considered present in the soil if it occurred at any abundance in any of the 202 unrarefied soil samples over two years. **b** The fractions of the phyllosphere communities present in the soil were even greater when considering the relative abundances of taxa; each circle represents the mean total relative abundance of leaf taxa present in the soil in up to eight replicate phyllosphere samples. **c** The relative abundances of taxa in pooled phyllosphere samples and pooled soil samples were positively correlated among taxa that were present at greater than 0.01% total abundance in the soil. Each black circle represents an OTU present in both phyllosphere and soil communities; a LOESS smoothing function is shown as a red line. Core members are shown as large green circles. **d**–**f** Source-sink models of phyllosphere community assembly from soils. Violin plots show the numbers of observed taxa in the phyllosphere were consistently lower than the richness values predicted by model simulations. The model assumed random increases and decreases in taxon abundances between time points and random immigration from the soil community (see Methods). Each circle represents a single phyllosphere sample

First, we interrogated the 2016 time series to determine the influence of soil-detected taxa on leaf microbial communities for both crops. As expected, the structures of leaf communities were highly distinct from soils (Fig. 2b, Supplementary Fig. 2B, Table 1). Though soil communities also changed seasonally, they experienced less overall change than the phyllosphere (Table 1, Fig. 2b, Supplementary Fig. 2C). While fertilization had no impact on phyllosphere communities, it did have small but significant influence on soil communities (Table 1). These seasonal and fertilization treatment patterns were reproduced in 2017 for switchgrass (Table 1).

To better understand the relationship between soil and phyllosphere communities, including the influence of rare members of the soil, we searched for phyllosphere taxa within the full soil dataset (not subsampled). Approximately 90% of phyllosphere OTUs were present in soil samples, with negligible differences between the two crops and modest variability over time (Fig. 3a). When considering the relative abundances of taxa, the mean fraction of phyllosphere communities found in soil samples was even higher at 98% and exhibited no clear trend over time (Fig. 3b). Our results show that the majority of abundant, commonly detected taxa in the phyllosphere are also present in the soil—albeit often at very low abundances (see below)—highlighting a potentially important role of the soil in harboring phyllosphere taxa between plant colonization events.

Given the large proportion of phyllosphere taxa present in the soil, we explored the potential role that immigration from the soil may have in shaping phyllosphere community composition and seasonal patterns. On balance, many abundant and persistent phyllosphere taxa were in low abundances in the soil, though there was a positive association between soil and leaf abundances for soil taxa > 1 e04 relative abundance (Fig. 3c). This result indirectly supports the presence of an ecological filter operating on the phyllosphere that favors some taxa while disfavoring others. To further investigate how ecological filtering may vary over the growing season, we compared observed trends in OTU richness with those predicted by a source-sink null model which simulated demographic stochasticity and random immigration from the soil between subsequent sampling points (see "Methods", Fig. 3d–f). Observed phyllosphere communities were dramatically less rich than null model predictions, again supporting the presence of a strong ecological filter. Such a filter could be due to host plant selection, environmental filtering, competitive exclusion among microbial taxa, or a combination of all three. According to this model, the strength of filtering did not trend consistently over the growing season.

Finally, other studies have found that soil microbes contribute more to early season phyllosphere communities[38], and we observed similar patterns: the most abundant soil taxa that were also detected on leaves were more prominent in the early season

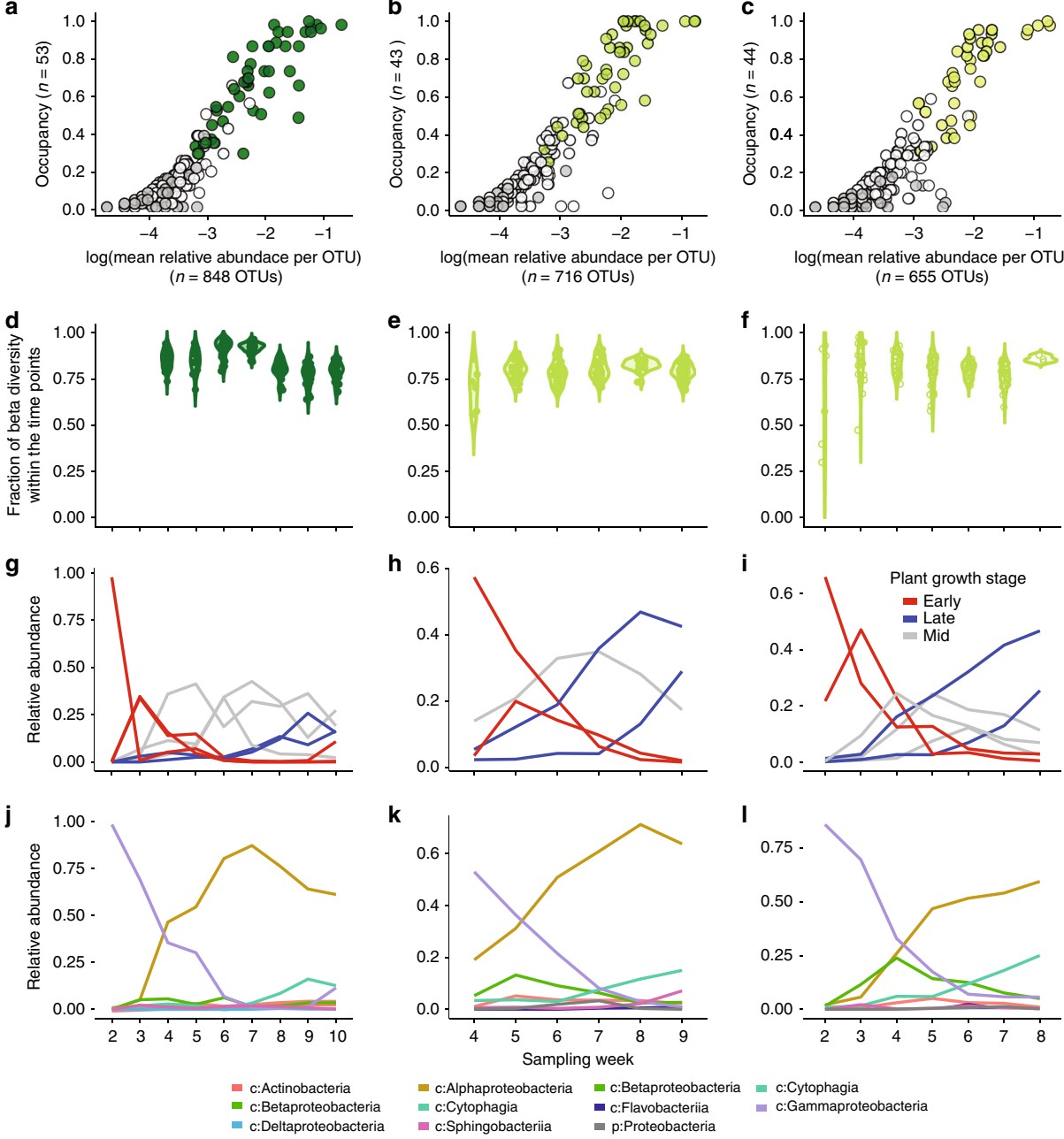

**Fig. 4** Selection and dynamics of core phyllosphere members. Abundance-occupancy of leaf taxa for **a** miscanthus 2016, **b** switchgrass 2016, and **c** switchgrass 2017, and their inclusion in their respective cores. Each point is an OTU. Abundance-occupancy distributions were calculated at each time point, and taxa that had 100% occupancy at any time point (e.g., were detected in all replicate plots at one sampling date) were included in the core (green filled circles). Non-core taxa that were detected in both crops (white/open circles), and crop-specific taxa (gray) are also indicated. **d–f** Contributions of the core taxa to changes in beta diversity over time. **g–i** Patterns of core taxa that share similar temporal changes, as determined by hierarchical clustering of standardized dynamics. Red lines are taxa that have early peaks in relative abundance, blue lines are late, and gray lines are mid-season. Colors correspond to the dendrograms in Fig. S7. **j–l** Patterns of core taxa summed by relative abundances within bacterial class. c: is class and p: is phylum, and lines are colored by taxonomic group

and then became rare and transient on leaves in the late season (Supplementary Fig. 4).

**Core members of the switchgrass and miscanthus phyllosphere.** There was high overlap between switchgrass and miscanthus phyllosphere communities and a trend towards increased intra-crop similarity during senescence (Supplementary Fig. 5). There was also a modest influence of host crop in 2016 (Table 1). Therefore, we defined a core microbiome for each crop and

season (Supplementary Data 1). We applied an established macroecological approach[39] to consider both the occupancy and abundance patterns of these taxa (Fig. 4a–c); abundance-occupancy relationships have been previously explored for microbial communities[40,41] and we utilize it here for ecologically informing a core microbiome. Occupancy is an ecological term that considers how consistently a taxon is detected across samples in the dataset, expressed as a proportion of occurrences given the total samples collected (e.g., 1.0 or 100%). Occupancy provides a dataset-aggregated term describing taxon persistence, which can

be informative for defining core taxa when datasets include a time series[42].

In contrast to the taxa unique to crops and years, which were rare and not persistent, most of the highly abundant and prevalent taxa were shared (Fig. 4a–c). We first quantified the abundance and occupancy distributions of OTUs, and then identified OTUs that were consistently detected across replicate plots at one sampling time (occupancy of 1) to include in the core. We found that these 44, 51 and 42 core taxa (as highlighted in Fig. 4a–c) contribute 84.4%, 79.5% and 79.4% to the total beta diversity in miscanthus 2016, switchgrass 2016 and 2017, respectively (Fig. 4d–f). While these core taxa were highly abundant and persistent on these crops' leaves, their functions are yet unknown and additional members could also transiently contribute. However, we suggest that the core taxa identified here should be prioritized for follow-up study of functionality and potential plant benefits.

Notably, if we had defined the core as those taxa uniquely detected on each crop (as in a Venn diagram analysis, Supplementary Fig. 6), we would have instead identified rare and transient taxa (Fig. 4a–c). Thus, a core analysis based on presence and absence, instead of on abundance and occupancy, would have provided a different, and arguably less ecologically relevant, core for these perennial crops. The approach used here provides a reproducible and conservative option for longitudinal series, and allows for systematic discovery[43,44] of a replicated core over time.

The core taxa included several Proteobacteria (*Methylobacterium*, *Sphingomonas*, and *Pseudomonas* spp.) and Bacteroidetes (*Hymenobacter* spp.). The taxonomic affiliations of these core taxa are consistent with the literature for other phyllosphere communities[2,3,18,35,45,46], providing new support for their seasonal importance in the phyllosphere.

We then performed a hierarchical clustering analysis of standardized (e.g., z-score) dynamics to explore seasonal trends of core taxa (Fig. 4g–i, Supplementary Fig. 7)[47]. This analysis identified several discrete, seasonally-defined groups of core taxa in switchgrass and miscanthus, respectively. Seasonal groups were taxonomically consistent across crops and years (Fig. 4j–l). This finding suggests potential for functional redundancy because closely related taxa are hypothesized to have substantial overlap in their functional repertoire[48]. The early-season groups (Supplementary Fig. 7, Fig. 4g–i red traces) included several Gammaproteobacteria (Fig. 4j–l). The late-season groups (Supplementary Fig. 7, Fig. 4g–i blue traces) were comprised of Alphaproteobacteria, Cytophagia and Actinobacteria and were pronounced in switchgrass (Fig. 4j–l). The third groups (Supplementary Fig. 7, Fig. 4g–i gray traces) included taxa that peaked in relative abundance mid-season, including Alphaproteobacteria and few taxa belonging to Beta- and Gammaproteobacteria, Cytophagia, Sphingobacteria, and Actinobacteria (Fig. 4j–l).

In addition, several Proteobacteria classes exhibited what appear to be compensatory dynamics (Fig. 5a). However, despite similarity in the membership and dynamics of the core microbiota on both crop plants, there were differences in the relative abundances of the same taxa across crops (Fig. 5b).

**Contributions of abiotic variables, space, time, and crop.** We summarize our analyses of the contributions of crop (host plant), space, time, and abiotic variables to the assembly of the core phyllosphere community in order of least to most important. There was no explanatory value of the spatial distance between the plots for community beta diversity (assessed by distance-decay of beta diversity using a Mantel test with a spatial distance, $r$: 0.013,

$p = 0.256$). This finding is different from a recent study of annual crops (common beans, canola, and soybean) that showed an influenced of sampling location on leaf microbiome structure[38]. Here, among those that variables were significant, crop (switchgrass or miscanthus) had the lowest explanatory value (Table 1). However, our work agrees with previous research that has shown a relationship between plant species/genotype and the leaf microbiota of perennial plants such as wild mustard[49], sugar cane[37], and tree species like birch, maple, and pine[50]. Time and measured abiotic factors had highest explanatory value (Table 1, Supplementary Table 3). Related, Copeland et al. 2015 showed that stage in plant development can influence leaf microbiome structure in annual crops.

## Discussion

We found that soil is a major reservoir of leaf microorganisms for switchgrass and miscanthus. Deep sequencing was required of the soils to observe many of the prominent leaf taxa for these perennial crops. This is in contrast to the studies of other plants that have suggested that the phyllosphere is comprised largely of passively dispersed and stochastically assembled microbes from the atmosphere[51–53], but in agreement with studies in sugarcane[37,54], grape[55], and perennial mustard[49] that have suggested soil origins of leaf microbiota. Notably, our analysis cannot inform directionality or mechanism of dispersal, which could have occurred between soil and leaf via wind, insects, or through grass emergence, etc. While 133 leaf OTUs (9%) observed in the phyllosphere could not be detected in the soil (using the unrarified soil dataset) and may be attributable to non-soil reservoirs, the vast majority of leaf microbes were detectable in local soils and non-neutral assembly patterns suggest both determinism and habitat filtering.

Our data suggest a compensatory relationship between members within the Proteobacteria, where members of Gammaproteobacteria and Alphaproteobacteria replace one another over time (Fig. 5). Such community transitions have been observed on the phyllosphere of crops such as sugarcane[37], common beans, soybeans, and canola[38]. A study of endophytic bacteria of prairie grasses, including switchgrass, showed the same trend in abundance of Gamma- and Alphaproteobacteria[56] suggesting that these phyllosphere taxa are facultative endophytes or are similarly affected by the plant development. The benefits plants may gain from these taxa are well characterized (see review from Bringel[57]), however it remains unknown what drives the exclusion of *Pseudomonas* and gives rise to Alphaproteobacteria (predominantly *Methylobacteria*) in the phyllopshere and endosphere. One possible explanation would be nutrient availability regulated by the plant development which would selectively influence the abundances of these taxa. Delmotte and colleagues[58] hypothesized that *Pseudomonas* specialize on monosaccharides, disaccharides and amino acids, whereas *Sphingomonas* and *Methylobacteria* are generalist scavengers that can subsist on a variety of substrates present at low amounts.

The crop-specific dynamics of some core taxa suggest their adaptation to or selectivity by the host plant. There were some core OTUs that had consistent dynamics across both crops, but these examples demonstrate, first, that dynamics can be crop specific, and second, that abiotic filtering of an OTU to a particular crop could be manifested as differences in dynamics in addition to the more extreme scenarios of taxon exclusion or crop specificity. Indeed, because the crop-unique taxa were generally rare and transient (Fig. 4), the dynamics of core taxa together with crop-distinctive dynamics may harbor clues as to the competitive landscape and microbially important changes in the hosts' leaf environments across crops.

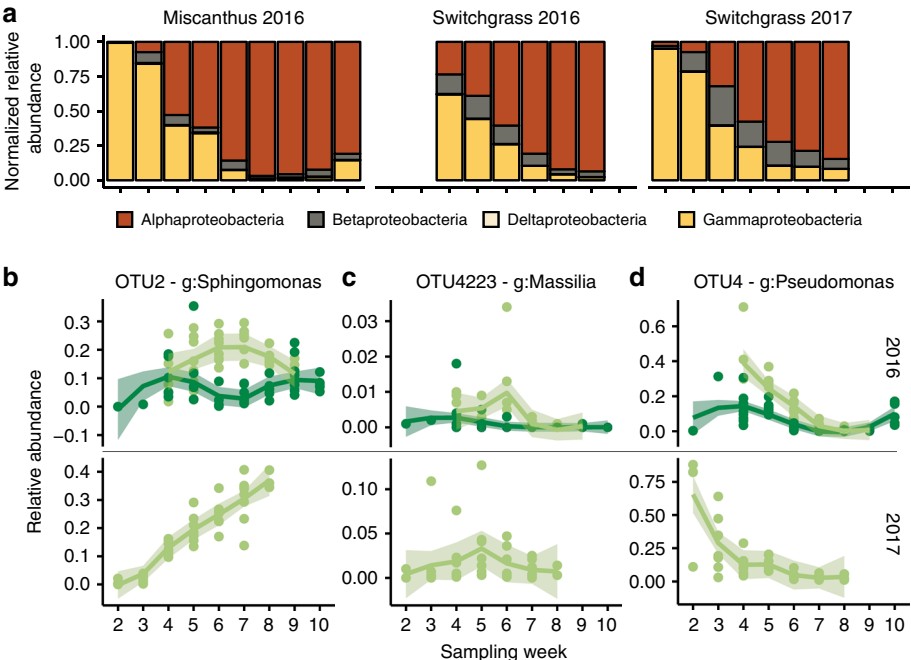

**Fig. 5** Compensatory patterns of Protobacteria classes over crops and season in the phyllosphere of switchgrass and miscanthus. Proteobacteria OTUs contributed 35.2% of the total taxa detected in the phyllosphere and contributed 116,760 total reads (34.1% of the leaf reads). **a** Changes in the relative contributions of all 521 leaf-detected Proteobacteria OTUs by class, over time. Colors are Proteobacteria class: Alphaproteobacterial are red, Betaproteobacteria are brown, Deltaproteobacteria are tan, and Gammaproteobacterial are yellow. **b**–**d** Vignettes showing different dynamics of Proteobacteria OTUs that were detected within the phyllosphere core microbiome, over time (upper is 2016 and lower is 2017) and across crops (miscanthus are dark green lines, switchgrass are light green lines). Genus is g, and shading around each line shows the 95% confidence interval around the replicate series

As a point of discussion, we also considered the potential role of priority effects on the community assembly. Priority effects occur when differences in colonization history result in different community assembly outcomes[59]. While we cannot claim that colonization histories had no effect on phyllosphere successional patterns, the data shown in Supplementary Figs. 3 and 5 suggest that any such effects were relatively minor. Specifically, beta-dispersion among plants of the same species decreased over time (Supplementary Fig. 3), and Bray-Curtis dissimilarity between switchgrass and miscanthus cohorts decreased over time (Supplementary Fig. 5). Both lines of evidence point to phyllosphere community convergence over time, despite early some differences in community composition, suggesting that colonization history per se had little effect on successional trajectories. This is not to say definitively that early colonists did not modify the phyllosphere environment such that particular species were subsequently favored or disfavored, but that early colonists did not modify the phyllosphere environment differentially and/or to a meaningful degree. The majority of the early colonizers were rare or absent by the end of the growing season (Fig. 4g–i), and so they did not modify the phyllosphere environment (i.e., exert priority effects) to ensure and/or enhance their competitive dominance (i.e., niche construction, or niche preemption).

To conclude, we investigated the assembly and seasonal dynamics of the phyllosphere and soil microbes of two perennial grasses, switchgrass and miscanthus, and found consistent community trajectories and memberships across growing seasons, suggesting that their key players are predictable and that most of them can be detected in associated soils. Understanding the seasonal patterns of these key taxa could be used to improve biomass production, plant health, or facilitate conversion. As seen in work by Agler et al.[60], the introduction or control of a few key microbial species can have significant impact on the host plant phenotype.

Next steps should be to interrogate core members for functionality and direct interactions with the plant, including investigations of the interactions among core members and with the host crop. This exploration lays the foundation for an approach to biofuel grass production that incorporates an understanding of host-microbe and microbe-microbe interactions.

## Methods

**Site description and sampling scheme.** Our study system is located within the Great Lakes Bioenergy Research Center (GLBRC), BCSE in Hickory Corners, Michigan (42º23'41.6" N, 85º22'23.1" W). We collected samples from two biofuel crops within the BCSE, switchgrass (*Panicum virgatum L.* cultivar Cave-in-rock) and miscanthus (*Miscanthus* x *giganteus*). Both crops had been continuously grown since 2008, in replicate 30 × 40 m plots arrayed in a randomized complete block design. Within each plot, nitrogen-free (no fertilizer) subplots were maintained in the western-most 3 m of each plot. We sampled replicate plots 1–4 in both the main and the nitrogen-free subplots. We collected leaf and bulk soil samples every three weeks across the 2016 growing season, including bare soil in April through senescence in October and November. In total, we collected 152 soil samples (72 switchgrass and 80 miscanthus) and 136 leaf samples (64 switchgrass and 72 miscanthus). At each sampling time, leaves were collected and pooled at three flags along a standardized path within each plot. Leaves were removed from the plant stem using ethanol sterilized gloves, then stored in sterile whirl-pak bags until processing. Bulk soil cores (2 × 10 cm) were collected at the same three locations within a plot, sieved through 4 mm mesh, then pooled and stored in whirl-pak bags. All samples were kept on wet ice for transport, then stored at −80 °C.

Soil physico-chemical characteristics (pH, lime, P, K, Ca, Mg, organic matter, $NO_3$–N, $NH_4$–N, and percent moisture) were measured by the Michigan State University (MSU) Soil and Plant Nutrient Lab (East Lansing, MI, USA, http://www.spnl.msu.edu/) according to their standard protocols. From each plot, 10 switchgrass leaves or 5 miscanthus leaves were processed for leaf dry matter content according to[61]. Dried leaves were ground to a fine powder using a Sampletek 200 vial rotator and iron roll bars (Mavco Industries, Lincoln, NE, USA), then carbon and nitrogen were measured on an elemental analyzer (Costech ECS 4010; Costech Analytical Technologies Inc, Valencia, CA, USA). Weather data was collected from the MSU Weather Station Network, for the Kellogg Biological Station location (https://mawn.geo.msu.edu) for each sampling day, and plant height and soil temperature were measured on a per-plot basis.

**Nucleic acid extraction and sequencing.** Throughout, we use microbiome to refer to the bacterial and archaeal members as able to be assessed with 16S rRNA gene sequence analysis. Soil microbial DNA was extracted using a PowerSoil microbial DNA kit (MOBio Inc. Carlsbad, California, USA) according to manufacturer's instructions. Phyllosphere epiphytic DNA was extracted from intact leaves using a benzyl chloride liquid:liquid extraction, followed by an isopropanol precipitation[62], using ~5 g of leaves (5–10 switchgrass leaves, or a minimum of 2 miscanthus leaves). Metagenomic DNA from both soil and phyllosphere was quantified using a qubit 2.0 fluorometer (Invitrogen, Carlsbad, CA, USA), and DNA concentrations were normalized between all samples prior to sequencing. Paired-end amplicon sequencing was completed by the Department of Energy's Joint Genome Institute (JGI) using an Illumina MiSeq sequencer, and using the 16S-V4 primers 515F (5′-GTGCCAGCMGCCGCGGTAA- 3′) and 806R (5′-GGACTACHVGGGTW TCTAAT-3′)[63], according to the JGI's standard operating protocols, and incorporating plastid- and mitochondria-blocking peptide nucleic acids (mPNA: 5′-GG CAAGTGTTCTTCGGA-3′, pPNA: 5′-GGCTCAACCCTGGACAG-3′) to prevent co-amplification of host-derived 16S rRNA genes[64].

**Sequence quality control and defining taxa.** BBDuk (v 37.96) was used to remove contaminants and trim adaptor sequences from reads. Reads containing 1 or more'N' bases, having an average quality score of less than ten or less than 51 bases were removed. Common contaminants were removed with BBMap (v 37.96). Primers were trimmed using cutadapt (v1.17). Reads were merged, dereplicated, clustered into 97% identity with usearch (v10.0.240), and classified against version 123 of the Silva Database[65] using sintax[66]. All reads classified as mitochondria, chloroplast or unclassified were removed before the analysis. Additionally, reads from 4371 OTUs assigned only to the domain level were extracted and reclassified using SINA online aligner (https://www.arb-silva.de/aligner/)[65]. Six hundred and ninety-six unclassified reads subsequently were confirmed to be Bacteria could then be classified to more resolved taxonomic levels. Remaining reads were BLASTed against the entire NCBI nucleotide database and specifically against the switchgrass genome to check for non-specific binding, but no hits could be found. We also performed analyses using OTUs defined at 100% sequence identity (zOTUs) and found that overarching patterns of beta diversity were statistically indistinguishable from those observed using OTUs defined at 97% sequence identity.

**Alpha and beta diversity.** For alpha and beta diversity analyses, we performed analyses to datasets subsampled to the minimum observed quality-filtered reads per sample (141), as well as to 1000 reads per sample. We did this to enable comparison of the most complete time series to the most complete comparative view of diversity. We report richness as total number of OTUs clustered at 97% sequence identity. We used the protest function in the vegan package in R[67] to test for synchrony in patterns across crops and years. To calculate beta dispersion, we used the betadisper function in the vegan package in R[67], which is a multivariate analogue of Levene's test for homogeneity of variances. PERMANOVA was used to test hypothesis of beta diversity using adonis function in the vegan package in R[67].

Though we performed analyses using 97% sequence identity for the OTU definitions, we also checked that beta-diversity patterns were maintained if we have used a 100% sequence identity for clustering, and found them to be statistically indistinguishable by two separate tests (Correlation in a symmetric Procrustes rotation = 0.9075, *p*-value 0.001 on 999 permutations; Mantel correlation = 0.6966, *p*-value 0.001 on 999 permutations), as reported for other studies[68]. Therefore, the strong seasonal patterns observed are robust regardless of using 97% or 100% OTUs.

**Source-sink models and contributions of soil taxa to leaf communities.** Given that virtually all phyllosphere taxa were present in the soil, we evaluated the degree to which observed seasonal patterns in phyllosphere community composition could be explained with a null model. The null model assumed functional equivalence among taxa and random source-sink dynamics from the soil. For each crop, we simulated community dynamics between each pair of sequential samples. This involved: (1) calculating the total number of 0.1% incremental increases/decreases in OTU relative abundances observed between the two sampling points; (2) randomly and iteratively selecting OTUs, weighting their probability of selection by their relative abundances, to increase or decrease by 0.1% increments of relative abundance until reaching the total number of observed increases/decreases for the sample pair; (3) counting the number of immigrant OTUs which appeared only in the second sample of the pair; (4) randomly selecting OTUs in the simulated community to decrease by 0.1% increments until the total decrease equaled the observed number of arriving OTUs multiplied by their median initial abundance (0.1%); and then (5) randomly selecting, again weighting their probability of selection by their relative abundance, the predicted number of immigrant OTUs from the soil community, such that the final simulated community abundance was equal to 1000 sequences. The source soil community was generated by pooling all soil samples from both years. We used this process to simulate demographic stochasticity. In the model, it was assumed that 0.1% incremental changes in relative abundance realistically reflects phyllosphere population dynamics, and that an immigration event results in the initial relative abundance of an OTU at 0.1%. Importantly, even if these assumptions are imprecise, the simulation provides a

consistent baseline of community composition against which to compare observations over the growing season.

**Core taxa selection.** To infer the core phyllosphere taxa and prioritize them for further inquiry, we calculated the abundance-occupancy distributions of taxa, as established in macroecology (e.g., Shade et al.[40]). For each OTU, we calculated occupancy and mean relative abundance at each time point by crop and year. Only OTUs with occupancy of 100% (found in all samples at a particular time point) were prioritized as core members. Using this conservative threshold for occupancy, we included all OTUs that had strong temporal signatures; these taxa also were in high abundance and were persistent as indicated by their abundance-occupancy distributions. These core taxa also represent potentially important players in plant development, as they were detected at least at one time point in all sampled fields.

We quantified the explanatory value of the core members to community temporal dynamics using a previously published method of partitioning community dissimilarity:[69]

$$C = \frac{BC_{core}}{BC_{all}} \qquad (1)$$

where C is the relative contribution of community Bray Curtis (BC) dissimilarity attributed to the core OTUs.

To be consistent with all other analyses in this work, we used the 97% sequence identity to define OTUs in our analysis of the core microbiome. We reasoned that 97% OTU clusters would be more conservative relative to 100% OTU clusters, given that any sequencing errors, as well as sequences that had less than perfect match among multiple copies of the 16S rRNA gene from the same genome, would result in inflation and potential overestimation of the core. We provide the representative sequence of each core OTU in Supplementary Data 1 so that the data can be re-analyzed using different OTU definitions and compared to the core detected here.

**Hierarchical clustering.** To understand the seasonal abundance patterns of the core taxa we performed hierarchical clustering. We used a z-scored relative abundance matrix subset to contain only core taxa to generate a w complete linkage distance matrix using the R function hclust()[47]. Groups of core taxa with similar dynamics were defined from the dendrogram using the function cutree() in R with number of desired groups (k=) to be close to the number of sampling time points; 8 for miscanthus 2016, 5 for switchgrass 2016 and 7 for switchgrass 2017.

**Reporting summary.** Further information on research design is available in the Nature Research Reporting Summary linked to this article.

## Data availability

The datasets generated and/or analyzed during the current study are available in the Joint Genomes Institute Genome Portal (https://genome.jgi.doe.gov/portal/) with projects designated by year and sample type (Project ID 1139694 [https://genome.jgi.doe.gov/portal/pages/dynamicOrganismDownload.jsf?organism=SwiandphylliTags_FD], and Project ID 1139696 [https://genome.jgi.doe.gov/portal/SwiandmsoiliTags_FD/SwiandmsoiliTags_FD.download.html] for 2016 season phyllosphere and soil; and Project ID 1191516 [https://genome.jgi.doe.gov/portal/SwiandphyiTagsII_FD/SwiandphyiTagsII_FD.download.html] and Project ID 1191517 for 2017 [https://genome.jgi.doe.gov/portal/pages/dynamicOrganismDownload.jsf?organism=SwiandsoiiTagsII_FD] season phyllosphere and soil sequences, respectively).

## Code availability

Our sequence analyses and statistical workflows are available at https://github.com/ShadeLab/PAPER_GradySorensenStopnisek_NatComm_2019.

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

## Acknowledgements

We thank S.H. Lee, M. Sleda, S. Wu and M. Nunez for technical assistance in the field and laboratory. This material is based upon work supported by the Great Lakes Bioenergy Research Center, U.S. Department of Energy, Office of Science, Office of Biological and Environmental Research under Award Numbers DE-SC0018409 and DE-FC02–07ER64494, by the National Science Foundation Long-term Ecological Research Program (DEB 1637653) at the Kellogg Biological Station, and by Michigan State University AgBioResearch. The work conducted by the U.S. Department of Energy Joint Genome Institute, a DOE Office of Science User Facility, is supported under Contract No. DE-AC02-05CH11231. This work was supported in part by Michigan State University through computational resources provided by the Institute for Cyber-Enabled Research. N.S. acknowledges support from the Michigan State Plant Resilience Institute.

## Author contributions

A.S. designed the study. A.S., K.L.G., J.S. and N.S. conducted field work. K.L.G. executed lab work. J.S., N.S., J.G. and A.S. analyzed the data. All authors discussed and revised the manuscript.

## Additional information

**Competing interests:** The authors declare no competing interests.

