## [Peer Review File · Nature Communications]

Reviewers' comments:

Reviewer #1 (Remarks to the Author):

Review of Grady, Shade et al:

The authors contribute a 16S based study of bacterial and archaeal microbiota associated with one season of *Miscanthus* growth and two seasons of switchgrass (*Panicum*) growth in conjunction with coordinated soils. The premise is that this work will add novel beneficial information for crop wellness and productivity for sustainable biofuel production - although this whole concept is sort of dropped after it is mentioned in the introduction. The manuscript would be improved if this focus could be revisited in light of all of the excellent results.

There are a lot of very interesting observations in this manuscript that are partially diluted by an over sharing of the less significant elements. The figures should be fewer and more concise to highlight the unique and interesting findings. The amount of data generated (# of sequence reads should be better explained). It sets off alarms when libraries are rarified to 146 sequences without explaining and proving evidence for why this is suitable. The authors seem determined to prove that soil is the primary driver of phyllosphere microbiota without exploring all of the biotic factors they claim comprise the drivers of phyllosphere microbiota (soil, atmosphere, and host plant). They describe with seeming certainty only three drivers and then don't seem to connect the dots - or incorporate their own hypothesis when they encounter variance that is not explained by soil.

The work is excellent and with some added discussion and cleaner, more succinct figures - will make a great contribution to the phyllosphere literature. Added discussion and additional literature review is also needed to make the work a valuable contribution to plant biofuel research as well.

Line 25: There is no data in the manuscript to support the statement that the authors effectively described the origins of the phyllosphere microbiota they profile in this work.

Lines 30 to 32: There is no evidence in the manuscript to support the following statement: "We found abundant and persistent core leaf taxa that originated in soil but were adapted for life on the leaf, rather than vagabonds that randomly disperse from air or soil."

Suggesting that origin and genomic adaptation can be inferred because the same 16S OTUs were observed in two environments (soil and phyllosphere) gets far beyond the scope of inference the data generated for this study can support. It is not possible to know if the origin of the OTUs is soil or air - if the air is not sampled in conjunction with the soil and phyllosphere sampling.

"vagabonds" ? perhaps you could find a word that is less anthropogenic and homeless sounding here - like "transient"?

Line 32: Please explain more clearly how the seasonal and host specific assemblages suggest/predict functional relationships with the plants.

Line 34: What is the foundational knowledge advance? Please describe more clearly how it (and /or data generated from this work) will advance crop wellness and productivity for agricultural biofuels. (The authors begin to achieve this justification more descriptively in 47 to 50)

Lines 46 to 48: What potentially microbially driven physicochemical parameters of the senesced crop are influenced by phyllosphere microbiome and may contribute to good biofuel? How does microbiota of the living plant potentially contribute to higher quality biofuels? A few sentences describing this would strengthen the applied value of this work - i.e. better address the goal of: "advancing crop wellness and productivity for agricultural biofuels".

Lines 51 to 64: These sentences provide a general phyllosphere intro. It would strengthen the

manuscript to provide a phyllosphere introduction that moves from the general phyllo intro - to why plant phyllo microbes matter for biofuels.

Lines 67 to 69: What you are capable of contributing to the field using the data assembled here with regard to “quantified drivers, and “soil influence on phyllosphere assembly”. Please explain more clearly.

Lines 73 to 74: “Exhaustive sampling” should be described with details and numbers. To make readers understand the power of your data. It would help to know precisely how much sequencing was done on these environments.

Line 86: Please improve clarity by replacing the word “simple” with more ecologically descriptive word -i.e. consistent, low diversity..stable..

Lines 89-95: The authors describe that 146 sequencing reads effectively describe leaf diversity, and that they have performed deeper sequencing and it does not make huge difference. Reword “at first we were surprised” (Lines 88-89) and provide more details to support these statements. It is hard to figure out how much sequencing was done at all without going to the supplemental materials – this info should be readily accessible in the main text. It sounds suspicious to hear that 146 was sufficient – without evidence/analyses done at 1k. Please show that 146 provides same results as 1k. Deeper sequencing described as 500 reads? (Line: 304). Nowhere in the manuscript or main figures do we see or read the details of the thousands of reads that were generated for the deep sequencing that was actually done.

Lines 113 - 136: Very interesting phenological studies of soil and phyllosphere communities. Expand conclusions from these data and observations but avoid concluding that phyllosphere microbiota comes from soil and not air when air was not studied (Lines 137-138). Studies that describe shared air and phyllosphere microbiota are cited. Because the majority of microbes in this work were shared between soil and phyllosphere is not sufficient to establish origin any more than it is for the air and phyllosphere studies. To make this claim – all three environments would need to be studied together (as the authors posit in lines 107 to 112)

Lines 147 to 208: – Very interesting core microbiome discussion – Figure 4 does not convey this information effectively. Pick the most important observations discussed and limit figure to those circularized temporal relative abundances.

Lines 210-226: Very interesting discussion on drivers – Line 223: describes 46.2% of variance not explained by measured factors– which authors describe as “unmeasured abiotic factors of importance” when from 107 to 112 – the authors have already suggested that “all” the forces that impact the phyllosphere are captured by soil, plants and atmosphere...

Line 302: rarifying to 146 really small # – why not 1000? While 146 may be suitable for phyllosphere – (and a sentence/analysis proving that is needed), there is no way it is suitable for soil.

Line 304: – Do you mean reads were rarified to 500 for soil? Please clarify and also provide proof at minimum 1k. A lot of confidence in this work is regained when observing Figure S1. Find a way to incorporate those statistics - if not the figure itself in the main work.

Figure 1: What is the value of including the 1.9 and 2% variance PCoA - sort of dilutes the more interesting trends that were observed? Suggest a smaller figure with only A and C.

Figure 2: There is so much excellent work in this manuscript – please think about how to really

highlight that in the first 3 figures- keeping in mind your premises and hypotheses : 1) phyllosphere microbial ecology description provided here is novel and valuable for plant biofuels, 2) three key forces influence the microbiota of the phyllosphere: soil, atmosphere, plant – for authors descriptive focus is on soil influence on phyllo micro. This figure is fine but not exciting and could be said in a single sentence. Try to avoid including figures that could be clearly stated with one sentence.

Figure 3: Very difficult to understand what significant finding is being shared here.

Figure 4: To make this figure immediately accessible – please limit to no more than 3 or 4 taxa with significant or contrasting trends with regard to their temporal incidence and relative abundance.

Figure 5: Legend for this figure describes very basics. How are these networks important for microbiota of plant biofuels? How do they support the hypothesis of soil as the preliminary driver of phyllo microbiota?

Figure 6: This could be said in a single sentence – What is the value of describing these class level relative abundances in histograms? Why not describe key genera? Or at least Family? Or leave out figure entirely and replace with more interesting supplemental figures?

Supplementary figures:

S1 – would prefer to see this in manuscript over current Figure 2

S3 – So interesting – a brief discussion of the shared and unique taxonomy with figure added to main manuscript would be of interest to readers.

Reviewer #2 (Remarks to the Author):

In the manuscript entitled “Assembly and seasonality of core phyllosphere microbiota on perennial biofuel crops”, the authors provide a survey of leaf bacterial and archaeal communities of two plants: switchgrass and *Miscanthus* over time. They also provide an estimate of the contribution of soil bacterial communities in seeding the leaf bacterial communities. The survey of the bacterial communities was done with 16S rRNA gene sequencing. Samples were collected from the plants and soil every 3 weeks across 2016/2017 from pre-emergence to senescence. The time-series sampling in the study is quite remarkable and provides an unprecedented perspective of the temporal heterogeneity in leaf bacterial communities through the growth season. Although I am impressed by the work presented in the manuscript and don't doubt its relevance for the phyllosphere scientific community, many aspects of the manuscript raised concerns as to the robustness and relevance of the results and discussion in their present form.

My main concern with the manuscript is that the analyses are based on the rarefaction of the leaf bacterial communities at a very low threshold, a threshold that doesn't seem to be enough to capture the overall diversity in the communities (see rarefaction curves in supp.). Therefore, I would need more information (as requested in the methods commentary below) to be able to accurately evaluate the sensitivity/robustness of the results. Another option for the authors would be to completely avoid rarefaction and use alternatives such as variance stabilizing transformation (Love et al. 2014, DESeq2). In addition, the authors talk abundantly of a “core” community without ever describing what is the biological relevance of describing the hypothetical role of this core bacterial community. The method used to define such a core community is unknown to me and has no references attached to it, providing very little information to its robustness or biological relevance. The statistical analyses are extended and mostly appropriate. However, adding Tables on to describe the multiple PERMANOVAs that were run is necessary for the readers to evaluate

how the models were run and what was the relative significance of each variable.

Title

It would be more appropriate to put bacterial communities or bacteriome in your title instead of "microbiome" because you provide only a survey of the leaf bacterial communities. Microbiome encompasses all microscopic organisms inhabiting an environment. I suggest also reviewing the whole manuscript to switch from microbiome to bacterial communities, or state at the beginning of your manuscript that for the sake of simplicity, you will use microbiome to refer to the bacterial/archaeal part of the microbial community.

Abstract

Lines 22-23: In your definition of the phyllosphere, the aerial surfaces include the microbial communities of the stems and the flowers. However, most of the work on the phyllosphere separate the microbial communities of these two surfaces, defining phyllosphere as relating to the leaf surfaces and interior. It would be more appropriate to rephrase to stick to the phyllosphere definition as in the literature.

Lines 25-27: "Here, we characterized the origins...". This is an overstatement as you limited your exploration to the contribution of the soil in seeding the leaf bacterial communities. Please rephrase maybe by using "sources" instead.

Lines 30-32: It would be more appropriate here to use scientific terminology rather than terms that relate to human behavior ("vagabonds"). What you actually mean here is that there seem to be a higher chance of dispersion from the soil to the leaf than from stochastic colonization from the air microbial seed pool (which you actually explain much better at lines 100-102).

Lines 32-34: The alternative hypothesis could also be that the bacterial communities provide little functions or benefits to the plant but instead are selected by the local micro-abiotic and biotic conditions, demonstrating a deterministic process of assembly in the leaf bacterial communities.

Lines 34-36: It is hard to see how the confirmation that host and abiotic conditions have selective powers on the microbiome could "advance goals to leverage native microbiomes to promote crop wellness and productivity in the field...". What is implied here? That maintaining the core leaf bacterial communities of switchgrass and Miscanthus will increase or support their plant productivity? Protect them from pathogens? Give them a higher resistance and resilience to biotic and abiotic stresses? It is hard to follow the scientific justification behind this sentence.

Introduction

Lines 45-47: What are "microbial equivalent of mansions"? What would be trees if switchgrass are mansions? Could you rephrase to provide a more scientific description of the abundant foliage of these two plant species?

Lines 47-50: A reformulation of these lines could strengthen the conclusion of the abstract.

Lines 51-55: It would be a good place to actually include host-microbe interactions studies that have demonstrated the effect of these interactions on plant fitness during drought (i.e. Fitzpatrick et al. 2018, Santos-Medellín et al. 2017; see full references below.)

Lines 61-64: Please put references at the end of sentence.

Line 67: Please define what you consider a "core phyllosphere microbiome member".

Results and Discussion

Lines 73-74: I think the figure shows quite the contrary as to the "exhaustive" sampling of the phyllosphere communities, especially when rarefied at 146 sequences per sample.

Line 76: A Table is needed to provide the reader with the model's equation, each variable degree of freedom, the pseudo F, the R2 and the p-value. The pseudoF provides actually very little information unless compared with other pseudoF of the model, which is quite hard to do when they are scattered in the text. Also, very hard to appreciate if the design was balanced, PERMANOVAs are a great way to test for differences in community structure but they are only robust to heteroskedasticity between groups (which is almost the case all the time when analyzing microbial communities) only if the design is balanced. Throughout the results, I would like to see a table for each permanova either in the main manuscript or in supplementary results to provide support to the statistical method used and provide the readers with the means to assess the strength of each variable in explaining bacterial community structure.

Line 88-91: You mention that "... we were surprised that 146 reads could well-describe the leaf diversity (we had performed much deeper sequencing), but inclusion of additional reads did not alter analysis outcomes..." but here you actually only tested two levels: 146 and 500 reads, right? And the rarefaction curves in supp. actually suggests that you don't capture the leaf community diversity at all at this threshold.

Line 107: Please add main or major before "sources of microorganisms" as other biotic agents such as arthropods can contribute to plant microbial communities.

Line 121: Typo in first pseudoF

Lines 137-139: Saying that "We conclude from these results that soil is the most substantial reservoir of leaf microorganisms" is an overstatement as you have nothing to compare it to. Only if you had sampled the air community through filters and the vertical legacy of microbial communities through generation you could say that. Please rephrase to say that soil is a major contributor or something more like it.

Lines 149-151: An ordination is not a robust support for such a statement, your PERMANOVA is. Please rephrase. Again here, please report R2 and link it to a table reporting both the full model equation and statistics.

Line 157-165: It would be more interesting to provide a discussion of why these core members could have a "temporal importance in the phyllosphere", what are their roles?

Line 182: Typo in Proteobacteria.

Line 189: One set of parentheses is enough.

Lines 225-226: What about stochasticity in initial colonization load. What about microbe-microbe interactions on the leaf?

Lines 230-231: What does it mean "We considered the sources of the phyllosphere communities..."? And you can't provide support to this statement "... found that the associated soil is likely the primary reservoir for these taxa." because you haven't compared its relative contribution to any other reservoir. You can only say that it makes a major contribution to phyllosphere bacterial communities.

Methods

Lines 301-302: 146 reads per sample seems to be a pretty low number of read for characterizing such a diverse community. Especially looking at the rarefaction curves in supplementary files, it seems that the diversity plateau is not at all reach at 146 sequences. If accurate, this would be a very strong argument against any rarefaction at this level and you might have to exclude more samples that had very low total sequence counts. An alternative to rarefaction could be to use

variance stabilizing transformation as presented by the group of Dr. Holmes (McMurdie & Holmes 2014) and implemented in DESeq2 (Love et al. 2014).

Could you also provide the range of reads per sample for both soil and leaf samples respectively? Same for the range of number of OTUs you found per sample. Also, please provide the total number of sequences for each dataset (leaf and soil) you based your analyses on.

Lines 308-309: Did you test that your alpha-diversity data was normally distributed and that the residuals of your model were homoscedastic? If yes, please provide this information in the methods. If they are not, you should use a non-parametric test.

Line 309: Reference to protest function R package is missing.

Line 311: Reference to betadisper function R package is missing.

Line 313: Reference to adonis function R package is missing.

Line 314: Reference to vegan R package is missing.

Lines 317-318: One set of parentheses is enough.

Lines 319-321: Is this based on other work? If yes please cite. If no, please justify biologically why these criteria are appropriate to identify a core microbiome. And what is a core microbiome? Please define.

Lines 326-328: Is there a reference that goes with this technique? I am new to it and would appreciate to see its mechanics.

Line 335: Reference to hclust function R package is missing.

Figures

Figure 1: When reported in the methods/results, I would suggest to remove the rarefaction thresholds from the figure legend.

References

- Fitzpatrick, C. R., Copeland, J., Wang, P. W., Guttman, D. S., Kotanen, P. M., & Johnson, M. T. (2018). Assembly and ecological function of the root microbiome across angiosperm plant species. *Proceedings of the National Academy of Sciences*, 201717617.
- Love, M. I., Huber, W., & Anders, S. (2014). Moderated estimation of fold change and dispersion for RNA-seq data with DESeq2. *Genome biology*, 15(12), 550.
- McMurdie, P. J., & Holmes, S. (2014). Waste not, want not: why rarefying microbiome data is inadmissible. *PLoS computational biology*, 10(4), e1003531.
- Santos-Medellín, C., Edwards, J., Liechty, Z., Nguyen, B., & Sundaresan, V. (2017). Drought stress results in a compartment-specific restructuring of the rice root-associated microbiomes. *MBio*, 8(4), e00764-17.

Reviewer #3 (Remarks to the Author):

1st review of K.L. Grady et al. "Assembly and seasonality of core phyllosphere microbiota on perennial biofuel crops" for Nature Communications.

In this manuscript, the authors identify the core phyllosphere microbiome members for two perennial biofuel crops, miscanthus and switchgrass. Then, they quantified drivers of their seasonal dynamics from weather, plant, and soil data, and assessed the contributions of soil microbes to the phyllosphere assembly. They found that leaves greatly differed from soils, suggesting host selection or adaptation of the phyllosphere microbiome (through enrichment most likely), where the soil acts as a reservoir of leaf microorganisms for perennial crops. Also, they

identify the core microbiome for both hosts and they use variance partitioning to find the main environmental drivers of host assembly.

While the topic is of great interest, and the data are of good quality and good temporal resolution, I have several major issues with this work. My major concern with this paper is related to its very descriptive and speculative nature in terms of the hypothesized assembly dynamics and mechanisms. I am not convinced, and I think readers won't be either. The authors need to provide a much stronger support to their hypothesis. This means confronting and testing their hypothesis with model expectations. I provide details below on how they can do so. In addition, I have some concerns about the novelty and robustness of the results.

1. Novelty and conceptual advance. It is not clear to me the novelty of the work. There are good and novel aspects in the paper: the study system, temporal dynamics, and suggestion of soils as the main reservoir of phyllosphere microbiomes (although I have serious doubts about this interpretation, see my point 3 below). But I do not see any major conceptual advance, beyond analysing assembly processes and temporal structure, which has been done before with other host-associated microbiomes. The question I pose myself, and the authors, is: Is this novel enough to justify its publication in a high-calibre journal as Nature Communications?

2. Non-stochastic assembly processes. The authors claim that the seasonal dynamics and the accumulation of taxa over time are suggestive of non-stochastic assembly processes (e.g. in lines 98-104). To sustain this statement the authors need to perform simulations of stochastic assembly and compare their results with the results emerging from the stochastic null-models. There are different null models that the authors can use, with different constraints depending on the question asked. For the non-random species richness accumulation, the authors can use different techniques to detrend the data and to test how this alters the observed pattern. For the seasonal data, the authors can consider a common species pool with the empirical numerical abundance of each prokaryote and then subsample from this pool. In general, the authors tend to ascribe differences to host selection. For example, in lines 197-199 the authors claim that differences in relative abundances "of the same taxa across plant hosts, suggesting microbiome selectivity for or by the host plant". That's only a possibility. But a more parsimonious explanation that the authors should check for by using null models is that small initial differences between the microbes colonizing each host plant, followed by strong priority effects (see e.g. Sprockett et al. 2018 Nature Rev. Gastro. And Hepat. 15: 197, for the human gut microbiota), can explain observed differences without any selectivity by the plant host.

3. Soils as the most substantial reservoir of leaf microorganisms. That's a really novel aspect that challenges prevailing wisdom. However, I am not convinced by the proof provided. The authors do not present data from aerial microbes. The fact that leaf microbes are also present in the soil, but at different abundances, could also be interpreted as if wind transports microbes that then do grow in abundance better in the leaf or in the soil. Hence, stochastic "seedling" from the atmosphere microbes, followed by species sorting (or habitat filtering), could lead to the observed pattern. Again, the way to test for their hypothetical "soil reservoir" hypothesis is to run the null models described above. Even better, the authors could model their hypothesis, using a source-sink metacommunity model, where the soil is the source, and leafs are the sink, plus some simple logistic population growth (or Gompertz model). They could then compare the results of this model with the outcome of an additional stochastic model that tests the "atmosphere reservoir" hypothesis. Then, they can answer the question: which of these two contrasting models better fit the data?

4. Core phyllosphere taxa definition. Core taxa are defined by a threshold for persistence and abundance (lines 151-156). Both thresholds seem arbitrary. Some authors have shown that the criteria for core definition require a systematic exploration (e.g., Astudillo et al. 2017, Env. Microb. 19: 1450). Different inclusion criteria (thresholds) can result in different patterns. The question then is: how robust are the patterns displayed by the core, the core microbial networks, and the

drivers of phyllosphere assembly, to different inclusion criteria?

Minor

- Line 88, and line 306: why this rarefaction cut-off? I assume it is linked to the minimum number of reads per sample in the database. This needs to be confirmed- as the authors say, it is a surprisingly low cut-off, although the relatively low richness of the system can explain this. The authors claim that the results of community structure are consistent between rarefactions to 146 and 500 reads. What aspects of community structure? If so, why not focusing on the rarefactions to 500 reads, providing a larger coverage?

- Line 216: Please provide references to the variance partitioning method used.

- Line 220-221: It is not clear whether the authors tested explicitly for spatial autocorrelation. They only mention "spatial distance between the plots had no explanatory value". To me, that only means that they have no distance-decay patterns, but not that spatial autocorrelation is affecting their results within this section.

>>>Thank you to the reviewers for their thorough and constructive comments on the previous version of the manuscript! We are pleased to report that we have addressed all major points in full and also have made the large majority of the suggested changes. Each reviewer query has been numbered and these numbers are used to link similar reviewer comments and author responses. Author responses are indented and indicated also with >>>. Line numbers refer to the numbers in the tracked changes (marked up) document.

Reviewers' comments:

Reviewer #1 (Remarks to the Author):

Review of Grady, Shade et al:

R1.1

The authors contribute a 16S based study of bacterial and archaeal microbiota associated with one season of Miscanthus growth and two seasons of switchgrass (Panicum) growth in conjunction with coordinated soils. The premise is that this work will add novel beneficial information for crop wellness and productivity for sustainable biofuel production - although this whole concept is sort of dropped after it is mentioned in the introduction. The manuscript would be improved if this focus could be revisited in light of all of the excellent results.

>>>First, thank you for your thorough and helpful review of the work, and we're glad that you found the work to have "excellent results" that will add "novel" information. We have taken your specific suggestions below and revised throughout to improve focus.

R1.2

There are a lot of very interesting observations in this manuscript that are partially diluted by an over sharing of the less significant elements. The figures should be fewer and more concise to highlight the unique and interesting findings. The amount of data generated (# of sequence reads should be better explained). It sets off alarms when libraries are rarified to 146 sequences without explaining and proving evidence for why this is suitable. The authors seem determined to prove that soil is the primary driver of phyllosphere microbiota without exploring all of the biotic factors they claim comprise the drivers of phyllosphere microbiota (soil, atmosphere, and host plant). They describe with seeming certainty only three drivers and then don't seem to connect the dots - or incorporate their own hypothesis when they encounter variance that is not explained by soil.

>>>Thank you for this comment, and we agree that the paper is meaty and some of the key elements may have gotten lost in overshared details. Therefore, we have edited the manuscript throughout to be more clearly focused on our two main research questions, which we now pose at the end of the end of the introduction: 1) Are there seasonal patterns of phyllosphere microbiome assembly? If so, are these patterns consistent across fields of the same crop, different crops, and years? 2) To what extent might soil serve as a reservoir of phyllosphere diversity?

>>Similarly, we have removed or moved less significant elements to supporting materials. We have also removed some redundant analyses (e.g. variance partitioning with PERMANOVA), see specific points below.

R1.3

The work is excellent and with some added discussion and cleaner, more succinct figures - will make a great contribution to the phyllosphere literature. Added discussion and additional literature review is also needed to make the work a valuable contribution to plant biofuel research as well.

>>>Thank you for the comment that the work is excellent and will make a great contribution that will be valuable. We have added discussion and literature, as suggested here.

Bodenhausen, N., Horton, M. W. & Bergelson, J. Bacterial Communities Associated with the Leaves and the Roots of *Arabidopsis thaliana*. *PLoS One* **8**, (2013).

Lindow, S. E. & Brandl, M. T. Microbiology of the phyllosphere. *Appl. Environ. Microbiol.* **69**, 1875–1883 (2003).

Barret, M. *et al.* Emergence shapes the structure of the seed microbiota. *Appl. Environ. Microbiol.* **81**, 1257–1266 (2015).

Hamonts, K. *et al.* Field study reveals core plant microbiota and relative importance of their drivers. *Environ. Microbiol.* **20**, 124–140 (2018).

Shade, A. *et al.* Macroecology to unite all life, large and small. *Trends Ecol. Evol.* (2018).
doi:<https://doi.org/10.1016/j.tree.2018.08.005>

Burns, A. R. *et al.* Contribution of neutral processes to the assembly of gut microbial communities in the zebrafish over host development. *ISME J.* **10**, 655–664 (2016).

Astudillo-García, C. *et al.* Evaluating the core microbiota in complex communities: A systematic investigation. *Environ. Microbiol.* **19**, 1450–1462 (2017).

Shade, A. *et al.* Culturing captures members of the soil rare biosphere. *Environmental Microbiology* **14**, 2247–2252 (2012).

Agler, M. T. *et al.* Microbial Hub Taxa Link Host and Abiotic Factors to Plant Microbiome Variation. *PLoS Biol.* **14**, 1–31 (2016).

R1.4

Line 25: There is no data in the manuscript to support the statement that the authors effectively described the origins of the phyllosphere microbiota they profile in this work.

>>>Thank you for this comment, we have replaced the word “origins” with the more appropriate term “reservoir” in this sentence and throughout, as we agree with the reviewer that using the word “origins” is an overstep of what the data can provide. We have also added new analyses, at the requests of R2 and R3, to quantify the contribution of soil taxa to the leaf habitat. This includes a comparing observed data against a source-sink null model and clarifying the contribution of soil taxa to the leaf environment (new **Figure 3**).

R1.5

Lines 30 to 32: There is no evidence in the manuscript to support the following statement: “We found abundant and persistent core leaf taxa that originated in soil but were adapted for life on the leaf, rather than vagabonds that randomly disperse from air or soil.”

Suggesting that origin and genomic adaptation can be inferred because the same 16S OTUs were observed in two environments (soil and phyllosphere) gets far beyond the scope of inference the data generated for this study can support. It is not possible to know if the origin of the OTUs is soil or air - if the air is not sampled in conjunction with the soil and phyllosphere sampling.

>>>Thank you for this thoughtful comment. We agree that the word “originated” is too strong and have tempered this and statements throughout with the word “reservoir”, which is a more neutral term that does not imply directionality. We agree that directionality and mechanism of arrival to the leaf is unknowable from the data (e.g., soil bacteria could have been dispersed to the leaf via the wind/air) and have added text acknowledging and discussing this (**L167**).

>>> We clarify and support with data that >90% of all phyllosphere taxa were detected in the local soils and that their contributions to the richness and abundances of the phyllosphere are very high (Figure 3A-B, Figure S4). Furthermore, we show that the contributions of taxa that were detected on the leaf but not detected in the soil made very small (~9%, **L169**), arguably negligible contributions to the total structure and richness, and also to the overarching patterns in beta diversity (see Figure 3C combined with Figure 4 of core taxa, which were all detected in the soil). Also, in response to a related comment from the other reviewers, we have also built source-sink models of leaf assembly to understand the potential contributions from the soil, and to evaluate whether seasonal phyllosphere dynamics could be driven merely by random dispersal from the soil (Figure 3D-F). Given the combined evidence of these analyses, we think that it is reasonable to suggest that soil is one of the major reservoirs of these leaf taxa.

R1.6

“vagabonds”? perhaps you could find a word that is less anthropogenic and homeless sounding here – like “transient”?

>>>Thank you, we have replaced this word with “transient” as suggested.

R1.7

Line 32: Please explain more clearly how the seasonal and host specific assemblages suggest/predict functional relationships with the plants.

>>>Removed “functional” from this line as it is an overinterpretation of our data .

R1.8

Line 34: What is the foundational knowledge advance? Please describe more clearly how it (and /or data generated from this work) will advance crop wellness and productivity for agricultural biofuels. (The authors begin to achieve this justification more descriptively in 47 to 50)

>>> Done (**L41-46, L57**).

R1.9

Lines 46 to 48: What potentially microbially driven physicochemical parameters of the senesced crop are influenced by phyllosphere microbiome and may contribute to good biofuel?
How does microbiota of the living plant potentially contribute to higher quality biofuels? A few sentences describing this would strengthen the applied value of this work – i.e. better address the goal of: “advancing crop wellness and productivity for agricultural biofuels”.

>>>We have shortened the abstract to meet requirements for the journal, but have added specific references in support of microbiome potential specific to biofuels (References 27-34, ~L57).

R1.10

Lines 51 to 64: These sentences provide a general phyllosphere intro. It would strengthen the manuscript to provide a phyllosphere introduction that moves from the general phyllo intro - to why plant phyllo microbes matter for biofuels.

>>>Thank you, we have moved this paragraph forward as suggested.

R1.11

Lines 67 to 69: What you are capable of contributing to the field using the data assembled here with regard to “quantified drivers, and “soil influence on phyllosphere assembly”. Please explain more clearly.

>>>Thank you for this comment, we have removed this sentence and replaced it with our two key questions, as suggested to focus the manuscript on the key results (L62).

R1.12

Lines 73 to 74: “Exhaustive sampling” should be described with details and numbers. To make readers understand the power of your data. It would help to know precisely how much sequencing was done on these environments.

>>>Thank you for this comment, and we agree. We have added an intro results paragraph to describe the sequencing effort and quality and ranges of quality reads per sample (see paragraph 1 of the Results “*Sequencing summary and alpha diversity*”), including reference to the rarefaction curves that have been moved from supporting to the main text (Figure 1A).

R1.13

Line 86: Please improve clarity by replacing the word “simple” with more ecologically descriptive word -i.e. consistent, low diversity..stable..

>>>We have changed “simple” to “Low richness” (L83)

R1.14

Lines 89-95: The authors describe that 146 sequencing reads effectively describe leaf diversity, and that they have performed deeper sequencing and it does not make huge difference. Reword “at first we were surprised” (Lines 88-89) and provide more details to support these statements.

It is hard to figure out how much sequencing was done at all without going to the supplemental materials – this info should be readily accessible in the main text. It sounds suspicious to hear that 146 was sufficient – without evidence/analyses done at 1k. Please show that 146 provides same results as 1k. Deeper sequencing described as 500 reads? (Line: 304). Nowhere in the manuscript or main figures do we see or read the details of the thousands of reads that were generated for the deep sequencing that was actually done.

>>>Thank you for this comment, and we agree that we could make it much easier for readers to find the relevant sequencing information quickly. Similar comments were made by the other reviewers. Therefore, we have subsampled to 1000 instead (at the loss of 19 switchgrass samples and 11 miscanthus samples, Table S1). We show that the seasonal patterns are the same regardless of 1000 or 141 sequences/sample, and we present the full time series in supporting material for comparison (Figure S1, S2, S3, Table S2). We have also removed the text “at first we were surprised” and added a new paragraph describing sequencing effort at details at the front of the Results section (L69).

>>>Note that while the previous version of the manuscript had a subsampling depth of 146 reads for the full time series, we now use a subsampling depth of 141 reads for the full time series. This is because we have omitted 5 OTUs that were unable to be taxonomically classified beyond Domain Bacteria. To be precise with the reads used for updated analysis in the supporting materials, we will refer to 141 reads throughout the response document in place of 146.

R1.15

Lines 113 - 136: Very interesting phenological studies of soil and phyllosphere communities. Expand conclusions from these data and observations but avoid concluding that phyllosphere microbiota comes from soil and not air when air was not studied (Lines 137-138).

Studies that describe shared air and phyllosphere microbiota are cited.

Because the majority of microbes in this work were shared between soil and phyllosphere is not sufficient to establish origin any more than it is for the air and phyllosphere studies. To make this claim – all three environments would need to be studied together (as the authors posit in lines 107 to 112)

>>>Thank you for this comment, which is similar to previous comments R1.5 and R1.4. Briefly, we agree and have removed the word “origin”, please see detailed response to the similar query R1.5.

R1.16

Lines 147 to 208: – Very interesting core microbiome discussion –
Figure 4 does not convey this information effectively. Pick the most important observations discussed and limit figure to those circularized temporal relative abundances.

>>>We’re glad that you think that the core microbiome discussion is interesting- Thank you!
>>>We have removed original Figure 4 and replaced with a new figure focusing on how we defined the core, the core members’ contributions to beta diversity, and their seasonal patterns. To new Figure 5B we have highlighted a few important observations of particular taxa as suggested.

R1.17

Lines 210-226: Very interesting discussion on drivers –
Line 223: describes 46.2% of variance not explained by measured factors– which authors describe as “unmeasured abiotic factors of importance” when from 107 to 112 – the authors have already suggested that “all” the forces that impact the phyllosphere are captured by soil, plants and atmosphere...

>>>Thank you for this comment. We realized that the variance partitioning was redundant with the PERMANOVA tests (for time, crop/host, Table 1) and EnvFit analysis for abiotic conditions (Table S3); we have removed it in an effort to streamline the manuscript to the most pertinent points (e.g., R1.2).

R1.18

Line 302: rarifying to 146 really small # – why not 1000? While 146 may be suitable for phyllosphere – (and a sentence/analysis proving that is needed), there is no way it is suitable for soil.

>>> We originally used this depth to maximize observation of the replicated time series for the leaf. At your and Reviewer 2’s suggestions, we have subsampled to 1000 instead of 146 (at the loss of 19 switchgrass samples and 11 miscanthus samples). We show that the seasonal patterns are the same regardless of 1000 or 141 sequences/sample, and we present the complete time series in supporting material for comparison (Figure S1, S2, S3, Table S2). For the soil-specific analyses of beta-diversity, we use a deeper sequencing depth of > 19K reads/sample.

R1.19

Line 304: – Do you mean reads were rarified to 500 for soil? Please clarify and also provide proof at minimum 1k. A lot of confidence in this work is regained when observing Figure S1. Find a way to incorporate those statistics - if not the figure itself in the main work.

>>>Thank you for your comment, which is similar to previous comments R1.14 and R1.18. In response, we have moved previous Figure S1 to the main text as new Figure 1. We also have provided a comparison between 141, 1000 and 10000 reads (Table S1, Table S2) and show that the overarching patterns of beta diversity (seasonality) are statistically consistent regardless of sequencing depth. We have also clarified the subsampling levels used for soils (~19K) in the Figure S2 legend and phyllosphere samples (1K) in the main text (**L76**) and figure legends.

R1.20

Figure 1: What is the value of including the 1.9 and 2% variance PCoA - sort of dilutes the more interesting trends that were observed? Suggest a smaller figure with only A and C.

>>>Done; we have simplified this figure as suggested. We have moved panel B including the soil only to supplementary information (Figure S2C). (Though we note that the soil PCoA, previously in B explains 13.2 and 7.1% variance on axes 1 and 2, respectively, and not 1.9 and 2% as the axes suggested – they were mislabeled for panel B only, and we apologize for the error.)

R1.21

Figure 2: There is so much excellent work in this manuscript – please think about how to really highlight that in the first 3 figures- keeping in mind your premises and hypotheses : 1) phyllosphere microbial ecology description provided here is novel and valuable for plant biofuels, 2) three key forces influence the microbiota of the phyllosphere: soil, atmosphere, plant – for authors descriptive focus is on soil influence on phyllo micro. This figure is fine but not exciting and could be said in a single sentence. Try to avoid including figures that could be clearly stated with one sentence.

>>>As suggested, we have moved the original Figure 2 of alpha diversity to supplemental (now Figure S1), and also removed the descriptive paragraph that originally accompanied the figure. We have kept panel D of original Figure 2 in the main text to demonstrate species accumulation non-linearly (now Figure 1B).

R1.22

Figure 3: Very difficult to understand what significant finding is being shared here.

>>>Thank you for this comment, we agree that this is a confusing figure. The main finding is that all of taxa detected in soil proportionally comprise most of the abundance in the phyllosphere, as well as most of the detected richness. We have replaced the original Figure 3 with a more compelling visual as suggested (new Figure 3AB). Figure 3C and Figure S4 show that, on balance, the taxa that are among the most abundant in the phyllosphere are relatively rare in the soil.

R1.23

Figure 4: To make this figure immediately accessible – please limit to no more than 3 or 4 taxa with significant or contrasting trends with regard to their temporal incidence and relative abundance.

>>>We agree; we have removed this figure and instead replaced it with the major temporal dynamics among the core taxa (new Figure 4) as well as a vignette, as suggested, of a few taxa with contrasting trends (Figure 5B).

R1.24

Figure 5: Legend for this figure describes very basics. How are these networks important for microbiota of plant biofuels? How do they support the hypothesis of soil as the preliminary driver of phyllo microbiota?

>>>Thank you for this comment. The network analysis would not address the contributions of soil to the phyllosphere. Instead, it showed potential relationships (negative and positive) among core phyllosphere taxa. We agree that the outcome of the network analyses – co-occurring taxa with similar seasonal dynamics - could be more concisely and compellingly visualized as a hierarchical cluster analysis. Therefore, we removed the networks and instead present these results (Figure 4G-I, Figure S8), which are identical in outcome but more digestible to the reader.

R1.25

Figure 6: This could be said in a single sentence – What is the value of describing these class level relative abundances in histograms? Why not describe key genera? Or at least Family? Or leave out figure entirely and replace with more interesting supplemental figures?

>>>Thank you, we have shortened it. These dynamics have been anecdotally described elsewhere, but never with this level observational effort within a seasonal series. At the reviewer's suggestion we have now included vignettes of particular taxa that demonstrate crop-specific dynamics (Figure 5).

Supplementary figures:

R1.26

S1 – would prefer to see this in manuscript over current Figure 2

>>>As suggested, this figure has been moved to the main text as Figure 1A and the original Figure 2 has been moved to supporting.

R1.27

S3 – So interesting – a brief discussion of the shared and unique taxonomy with figure added to main manuscript would be of interest to readers.

>>> Though we chose to keep Figure S3 in the supporting materials (now Figure S6), we note that the abundance-occupancy analysis (Figure 4A-C) for defining core taxa and related discussion includes these completely shared taxa (gray points) and the associated text also discusses taxonomy in detail.

>>>Thank you again for your thoughtful and thorough review of the work! Your suggestions have strengthened the piece and we are grateful to have had such positive and constructive reviewers for our work!

Reviewer #2 (Remarks to the Author):

R2.1

In the manuscript entitled “Assembly and seasonality of core phyllosphere microbiota on perennial biofuel crops”, the authors provide a survey of leaf bacterial and archaeal communities of two plants: switchgrass and *Miscanthus* over time. They also provide an estimate of the contribution of soil bacterial communities in seeding the leaf bacterial communities. The survey of the bacterial communities was done with 16S rRNA gene sequencing. Samples were collected from the plants and soil every 3 weeks across 2016/2017 from pre-emergence to senescence. The time-series sampling in the study is quite remarkable and provides an unprecedented perspective of the temporal heterogeneity in leaf bacterial communities through the growth season. Although I am impressed by the work presented in the manuscript and don't doubt its relevance for the phyllosphere scientific community, many aspects of the manuscript raised concerns as to the robustness and relevance of the results and discussion in their present form.

>>>Thank you for your thorough and thoughtful review; we're glad that you find that the study is “remarkable” and “provides an unprecedented perspective of temporal heterogeneity”. We think that the new subsampling depth and analyses we provide in the response document and substantially revised manuscript will alleviate your concerns.

R2.2

My main concern with the manuscript is that the analyses are based on the rarefaction of the leaf bacterial communities at a very low threshold, a threshold that doesn't seem to be enough to capture the overall diversity in the communities (see rarefaction curves in supp.). Therefore, I would need more information (as requested in the methods commentary below) to be able to accurately evaluate the sensitivity/robustness of the results. Another option for the authors would be to completely avoid rarefaction and use alternatives such as variance stabilizing transformation (Love et al. 2014, DESeq2).

>>>Thank you for your comment, which is similar to previous comments R1.14, R1.18, and R1.19. Please see responses below and also reference to the prior comments.

R2.3

In addition, the authors talk abundantly of a “core” community without ever describing what is the biological relevance of describing the hypothetical role of this core bacterial community. The method used to define such a core community is unknown to me and has no references attached to it, providing very little information to its robustness or biological relevance.

>>>Thank you for this comment, and we agree that we must do a better job at describing the biological relevance of the core. We have added this discussion (starting at **L178**) as suggested.

>>> For the method used to define the core, please be assured that we have not invented it and we apologize for not introducing the method sufficiently in the previous version of the manuscript. Rather, it is a standard analysis in macroecology to dually consider the complementary information of mean relative abundance and occupancy among taxa in a community, and we have recently written an article discussing its utility for microbial systems (Shade et al. 2017, see also Gaston et al. 2000). Abundance-occupancy distributions have been applied in the microbial ecology literature as the basis for neutral models in other labs as well as in ours (e.g., Sloan et al. 2007, Burns et al 2016, Lee and Sorensen et al. 2017). Briefly, the neutral abundance-occupancy expectation would be that very abundant taxa would occur in many samples, while very rare taxa would occur in few. In this work, we have used abundance-occupancy distribution to prioritize taxa in high occupancy across replicate time points. This follows ideas from our previous work in which we discuss how to define a core using persistence over time as well as abundance (Shade and Handelsman 2012).

>>>Also, the reviewer brings up a very good point that many microbial ecology readers may be unfamiliar with abundance-occupancy and its utility, and that we really needed to provide more information on it. Therefore, we have added text and references to motivate the analysis and provide some references to the relevant literature (**L178-187**). This method has an advantage over presence absence in providing ecological insight, and offer a brief comparison of the presence/absence approach in the new version (**L200-204**).

R2.4

The statistical analyses are extended and mostly appropriate. However, adding Tables on to describe the multiple PERMANOVAs that were run is necessary for the readers to evaluate how the models were run and what was the relative significance of each variable.

>>>Thank you for the positive comments about the analyses. At your suggestion, we have now provided all the full PERMANOVA in **Table 1**.

R2.5

Title

It would be more appropriate to put bacterial communities or bacteriome in your title instead of “microbiome” because you provide only a survey of the leaf bacterial communities. Microbiome encompasses all microscopic organisms inhabiting an environment. I suggest also reviewing the whole manuscript to switch from microbiome to bacterial communities, or state at the beginning of your manuscript that for the sake of simplicity, you will use microbiome to refer to the bacterial/archaeal part of the microbial community.

>>> Added clarification as suggested. We also refer to “bacterial and archaeal communities” in several other places in the manuscript (e.g., **L29** in abstract, **L62** in the introduction, **L100** in results , **L305** in methods)

R2.6

Abstract

Lines 22-23: In your definition of the phyllosphere, the aerial surfaces include the microbial communities of the stems and the flowers. However, most of the work on the phyllosphere separate the microbial communities of these two surfaces, defining phyllosphere as relating to the leaf surfaces and interior. It would be more appropriate to rephrase to stick to the phyllosphere definition as in the literature.

>>>Thank you for this comment. We have updated the definition to be the “aerial parts” (as per Vorholt 2012) and then defined that we are investigating the surfaces in the first occurrence (**L63, L70**)

R2.7

Lines 25-27: “Here, we characterized the origins...”. This is an overstatement as you limited your exploration to the contribution of the soil in seeding the leaf bacterial communities. Please rephrase maybe by using “sources” instead.

>>>Thank you for this comment, which is similar to that of Reviewer 1, and we agree that the term “origin” is an overstep. We have tempered throughout and replaced this word and clarified our meaning throughout to be the more neutral term “reservoir.”

R2.8

Lines 30-32: It would be more appropriate here to use scientific terminology rather than terms that relate to human behavior (“vagabonds”). What you actually mean here is that there seem to be a higher chance of dispersion from the soil to the leaf than from stochastic colonization from the air microbial seed pool (which you actually explain much better at lines 100-102).

>>>Thank you for this comment, which is similar to that of R1.6. We have replaced the term with “transient” at the suggestion of R1.

R2.9

Lines 32-34: The alternative hypothesis could also be that the bacterial communities provide little functions or benefits to the plant but instead are selected by the local micro-abiotic and biotic conditions, demonstrating a deterministic process of assembly in the leaf bacterial communities.

>>>We agree and have added such text to the discussion regarding habitat filtering/deterministic assembly (e.g., **L230, L237**).

R2.10

Lines 34-36: It is hard to see how the confirmation that host and abiotic conditions have selective powers on the microbiome could “advance goals to leverage native microbiomes to promote crop wellness and productivity in the field...”. What is implied here? That maintaining the core leaf bacterial communities of switchgrass and Miscanthus will increase or support their plant productivity? Protect them from pathogens? Give them a higher resistance and resilience to biotic and abiotic stresses? It is hard to follow the scientific justification behind this sentence.

>>>Thank you for this comment that is similar to R1. As this sentence is in the ending of the abstract, we have expanded on these thoughts by incorporating discussion and supporting references into other parts of the manuscript. Please see R1.3 for additional references incorporated into the discussion.

R2.11

Introduction

Lines 45-47: What are “microbial equivalent of mansions”? What would be trees if switchgrass are mansions? Could you rephrase to provide a more scientific description of the abundant foliage of these two plant species?

>>>Done. We have provided a more scientific description (**L54-56**)

R2.12

Lines 47-50: A reformulation of these lines could strengthen the conclusion of the abstract.

>>>We have reformulated the introduction according to recommendations by R1 to end with our two major scientific questions.

R2.13

Lines 51-55: It would be a good place to actually include host-microbe interactions studies that have demonstrated the effect of these interactions on plant fitness during drought (i.e. Fitzpatrick et al. 2018, Santos-Medellin et al. 2017; see full references below.)

>>>Thank you for the references – we have added them as suggested (near **L57**)

R2.14

Lines 61-64: Please put references at the end of sentence.

>>>In this case, the reference applied to the first clause and not the second, and so it would have been misplaced if we had put it at the end of the sentence. But, to follow the reviewer’s suggestion, we have reorganized this sentence so that the original first clause is now at the end of the sentence and the reference can be appropriately placed both with its correct clause and at the end of the sentence

Line 67: Please define what you consider a “core phyllosphere microbiome member”.

>>Done (**L32**, paragraph including **L177**)

Results and Discussion

R2.15

Lines 73-74: I think the figure shows quite the contrary as to the “exhaustive” sampling of the phyllosphere communities, especially when rarefied at 146 sequences per sample.

>>> Thank you, we are happy to address this comment. Please see prior detailed responses to: R.14 and R.18. Briefly, we have increased subsampling depth to 1000 sequences per sample and have also included the lower sampling depth to maintain the full time series, reporting this information for full transparency in supporting materials.

R2.16

Line 76: A Table is needed to provide the reader with the model’s equation, each variable degree of freedom, the pseudo F, the R2 and the p-value. The pseudoF provides actually very little information unless compared with other pseudoF of the model, which is quite hard to do when they are scattered in the text. Also, very hard to appreciate if the design was balanced, PERMANOVAs are a great way to test for differences in community structure but they are only robust to heteroskedasticity between groups (which is almost the case all the time when analyzing microbial communities) only if the design is balanced. Throughout the results, I would like to see a table for each permanova either in the main manuscript or in supplementary results to provide support to the

statistical method used and provide the readers with the means to assess the strength of each variable in explaining bacterial community structure.

>>> Thank you for the great suggestion to collate all of the PERMANOVA tests and outcomes. We have now provided all PERMANOVA results in Table 1.

>>>In regard to the heteroscedasticity, we have also used a permuted analysis of beta dispersion (PERMDISP, Anderson 2006) to test for differences in variability within and between groups (Figure S3, Figure S5). We are interested not only in differences in centroids but also in dispersion, which is an interesting ecological property that has gained popularity among those interested in dysbiosis (Zaneveld et al. 2017).

R2.17

Line 88-91: You mention that "... we were surprised that 146 reads could well-describe the leaf diversity (we had performed much deeper sequencing), but inclusion of additional reads did not alter analysis outcomes..." but here you actually only tested two levels: 146 and 500 reads, right? And the rarefaction curves in supp. actually suggests that you don't capture the leaf community diversity at all at this threshold.

>>> Thank you again for this comment. We have tested three levels (Table S1, 141, 1000 and 10,000 reads per sample) and assure the reviewer and show that the temporal patterns of beta diversity are upheld across subsampling depths. Please see previous responses (R1.14 and R1.18, R1.19) and also new Figure S1, Figure S2, Table S1 and Table S2. We capture the vast majority of the core taxa at the lowest subsampling depth and because these core taxa are driving the major patterns in beta diversity (Figure 4), and because the diversity on the leaf is low <150 taxa at any particular time, it is not inappropriate to use the lower subsampling depth to observe the full time series, as in Figure S2. However, for simplicity we show the patterns at a subsampling depth of 1000 at the cost of a reduced time series.

R2.18

Line 107: Please add main or major before "sources of microorganisms" as other biotic agents such as arthropods can contribute to plant microbial communities.

>>>Thank you, we have done as suggested and also added arthropod vectors (**L117**).

R2.19

Line 121: Typo in first pseudo

>>>We apologize for the typo. At the suggestion of R1 and related comments by you, we have removed this text in lieu of new PERMANOVA Table 1.

R2.20

Lines 137-139: Saying that "We conclude from these results that soil is the most substantial reservoir of leaf microorganisms" is an overstatement as you have nothing to compare it to. Only if you had sampled the air community through filters and the vertical legacy of microbial communities through generation you could say that. Please rephrase to say that soil is a major contributor or something more like it.

>>>Thank you for this comment We have tempered this statement as suggested, and also performed several additional analyses, including source-sink modeling suggested by Reviewers 1 and 3 to support the idea that soil is a major reservoir of these leaf taxa for switchgrass and miscanthus. However, we cannot comment as to their mechanism of arrival as our data cannot speak to that. Please also see prior response to R1.5, R1.15, R1.22 and R3.

R2.21

Lines 149-151: An ordination is not a robust support for such a statement, your PERMANOVA is. Please rephrase. Again here, please report R2 and link it to a table reporting both the full model equation and statistics.

>>>We have rephrased the statement and provided all PERMANOVA information in Table 1. We provide additional evidence for the crop separation mid-season (and then convergence again by senescence) by providing the Bray-Curtis dissimilarity between crops, at each time point (Figure S5).

R2.22

Line 157-165: It would be more interesting to provide a discussion of why these core members could have a “temporal importance in the phyllosphere”, what are their roles?

>>>Thank you for this question. This statement is aligned with our definition of a core according to its ecological distributions in abundance and occupancy. We think that the expansion of the discussion of the core regarding these distributions addresses this question. Please see more detailed, related response to R2.3.

R2.23

Line 182: Typo in Proteobacteria.

>>>Fixed.

R2.24

Line 189: One set of parentheses is enough.

>>>Thank you for this comment, and we apologize for the oversight. Our reference formatting software automatically adds parentheses after references, and so they must be manually removed when occurring inside an existing bracket.

R2.25

Lines 225-226: What about stochasticity in initial colonization load. What about microbe-microbe interactions on the leaf?

>>> At the request of reviewer 1 to better focus the work, we have omitted this ending paragraph, which was largely redundant with discussion in other parts of the work. Please see responses to Reviewer 3 re: priority effects in colonization. But, we agree that microbe-microbe interactions may be important (**L273**), and we would like to share that we are working on these interactions studies currently with members of this identified core microbiome that we have isolated from the environment.

R2.26

Lines 230-231: What does it mean “We considered the sources of the phyllosphere communities...”? And you can’t provide support to this statement “... found that the associated soil is likely the primary reservoir for these taxa.” because you haven’t compared its relative contribution to any other reservoir. You can only say that it makes a major contribution to phyllosphere bacterial communities.

>>>Thank you for this comment, and we understand its intent and confirm that we have tempered statements throughout about origins, replacing instead with the neutral term reservoir at the suggestion of Review 1. First, we have clarified the motivation and sentences starting in **L116**. In the revision, we can statistically assert from the source-sink models of assembly (as suggested by reviewer 3), that, given taxon overlap between the soil and the leaf and the observation of non-random assembly over time, that ~9% of the taxa ever detected on the leaf were never detected in the soil, while the remaining majority of OTUs were detected in the soil. So, while these other sources may have contributed (**L167**), their collective contribution would still be in the minority of the total leaf taxa detected in this study, in the soil. Please see detailed and related response to R2.3. We also add qualification that the mechanism of dispersal to the leaf is unknowable by these data (**L167**)

Methods

R2.27

Lines 301-302: 146 reads per sample seems to be a pretty low number of read for characterizing such a diverse community. Especially looking at the rarefaction curves in supplementary files, it seems that the diversity plateau is not at all reach at 146 sequences. If accurate, this would be a very strong argument against any rarefaction at this level and you might have to exclude more samples that had very low total sequence counts. An alternative to rarefaction could be to use variance stabilizing transformation as presented by the group of Dr. Holmes (McMurdie & Holmes 2014) and implemented in DESeq2 (Love et al. 2014).

>>>Thank you for this comment, which is identical also to your comment R2.20 and was addressed there.

R2.28

Could you also provide the range of reads per sample for both soil and leaf samples respectively? Same for the range of number of OTUs you found per sample. Also, please provide the total number of sequences for each dataset (leaf and soil) you based your analyses on.

>>>Absolutely! Please see Table S1 for these details. Also, we have added a new results paragraph describing the sequencing effort (See paragraph subsection: "Sequencing summary and alpha diversity").

R2.29

Lines 308-309: Did you test that your alpha-diversity data was normally distributed and that the residuals of your model were homoscedastic? If yes, please provide this information in the methods. If they are not, you should use a non-parametric test.

>>>Thank you for this comment. In an effort to clarify the work towards the key findings (and at the specific request of reviewer 1, general comment R1.2 and other minor comments of R1.), we have moved alpha diversity patterns to supporting materials and omitted the associated analyses.

R2.30

Line 309: Reference to protest function R package is missing.

>>>This function is in the vegan package in R (Okasanan 2015). We have added the reference and note that the annotated computing workflow also contains the package information.

R2.31

Line 311: Reference to betadisper function R package is missing.

>>>This function is in the vegan package in R (Okasanan 2015). We have added the reference and note that the annotated computing workflow also contains the package information.

R2.32

Line 313: Reference to adonis function R package is missing.

>>>This function is in the vegan package in R (Okasanan 2015). We have added the reference and note that the annotated computing workflow also contains the package information.

R2.33

Line 314: Reference to vegan R package is missing.

>>> We have added the reference (Okasanan 2015).

R.34

Lines 317-318: One set of parentheses is enough.

>>>Thank you for this comment, and we apologize for the oversight. Our reference formatting software automatically adds parentheses after references, and so they must be manually removed when occurring inside an existing bracket.

R2.35

Lines 319-321: Is this based on other work? If yes please cite. If no, please justify biologically why these criteria are appropriate to identify a core microbiome. And what is a core microbiome? Please define.

>>>Thank you for this comment, which is identical to previous comment R2.3. Please see our detailed response to that comment.

R2.36

Lines 326-328: Is there a reference that goes with this technique? I am new to it and would appreciate to see its mechanics.

>>> We have replaced the LSA network with a more straightforward and digestible hierarchical clustering analysis of standardized dynamics, at the request of Reviewer 1.

>>>But, for the curious: The local similarity analysis (LSA) was originally published for use with microbial fingerprinting time series from the BATS time series, from the Furhman lab at UCLA (Ruan et al. 2006), and was later extended and upgraded to a web version by (Xia et al. 2011). We like it for time series because it can incorporate a temporal lag as specified by the user. We also like it because it provides a linear (Pearson's) and non-linear (Local Similarity) score to explore for relationships.

R2.37

Line 335: Reference to hclust function R package is missing.

>>>hclust is part of the base stats available in R. The reference for R is: R Core Team (2018). R: A language and environment for statistical computing. R Foundation for Statistical Computing, Vienna, Austria. <https://www.R-project.org/>

Figures

R2.38

Figure 1: When reported in the methods/results, I would suggest to remove the rarefaction thresholds from the figure legend.

>>>Thank you for this comment, which is opposite some comments of R1.14 who asked for improved transparency in subsampling depth. Therefore, we have edited throughout to precisely state subsampling depth wherever appropriate in legends, methods, and results.

R2.39

References

Fitzpatrick, C. R., Copeland, J., Wang, P. W., Guttman, D. S., Kotanen, P. M., & Johnson, M. T. (2018). Assembly and ecological function of the root microbiome across angiosperm plant species. *Proceedings of the National Academy of Sciences*, 201717617.

Love, M. I., Huber, W., & Anders, S. (2014). Moderated estimation of fold change and dispersion for RNA-seq data with DESeq2. *Genome biology*, 15(12), 550.

McMurdie, P. J., & Holmes, S. (2014). Waste not, want not: why rarefying microbiome data is inadmissible. *PLoS computational biology*, 10(4), e1003531.

Santos-Medellín, C., Edwards, J., Liechty, Z., Nguyen, B., & Sundaresan, V. (2017). Drought stress results in a compartment-specific restructuring of the rice root-associated microbiomes. *MBio*, 8(4), e00764-17.

>>>Thank you for these references, we have incorporated all of them with the exception of Love et al. 2014 and McMurdie et al. 2014 because we did not use DESeq.

>>>Thank you again for your thoughtful and thorough review of the work! Your suggestions have strengthened the piece and we are grateful to have had such positive and constructive reviewers for our work!

Reviewer #3 (Remarks to the Author):

1st review of K.L. Grady et al. "Assembly and seasonality of core phyllosphere microbiota on perennial biofuel crops" for Nature Communications.

R3.0

In this manuscript, the authors identify the core phyllosphere microbiome members for two perennial biofuel crops, miscanthus and switchgrass. Then, they quantified drivers of their seasonal dynamics from weather, plant, and soil data, and assessed the contributions of soil microbes to the phyllosphere assembly. They found that leaf assembly greatly differed from soils, suggesting host selection or adaptation of the phyllosphere microbiome (through enrichment most likely), where the soil acts as a reservoir of leaf microorganisms for perennial crops. Also, they identify the core microbiome for both hosts and they use variance partitioning to find the main environmental drivers of host assembly.

While the topic is of great interest, and the data are of good quality and good temporal resolution, I have several major issues with this work. My major concern with this paper is related to its very descriptive and speculative nature in terms of the hypothesized assembly dynamics and mechanisms. I am not convinced, and I think readers won't be either. The authors need to provide a much stronger support to their hypothesis. This means confronting and testing their hypothesis with model expectations. I provide details below on how they can do so. In addition, I have some concerns about the novelty and robustness of the results.

>>>Thank you for your thorough review of the work and for your suggestions for improvement. At your suggestion, we have applied source-sink models to inform quantitative predictions of phyllosphere assembly, and in doing so we have brought on another author (John Guittar) with expertise in these models to assist. The new analysis and results can be found in Figure 3 and described in the Results subsection "*Contribution of soil microorganisms to phyllosphere assembly.*" Briefly, the results show that leaf assembly is deterministic early in the season and diverges from the expectation of random assembly from the soil, providing additional quantitative evidence in stronger support of our initial interpretation of the descriptive results.

R3.1

1. Novelty and conceptual advance. It is not clear to me the novelty of the work. There are good and novel aspects in the paper: the study system, temporal dynamics, and suggestion of soils as the main reservoir of phyllosphere microbiomes (although I have serious doubts about this interpretation, see my point 3 below). But I do not see any major conceptual advance, beyond analysing assembly processes and temporal structure, which has been done before with other host-associated microbiomes. The question I pose myself, and the authors, is: Is this novel enough to justify its publication in a high-calibre journal as Nature Communications?

>>>Thank you for this comment, and we appreciate your agreement about the novelty of the study system, characterization of temporal dynamics and suggestion of soils as a major reservoir of leaf taxa.

We note that reviewers 1 and 2 also agree that the time series is “unprecedented,” “excellent” and would provide “novel beneficial information”. We believe that, given the seasonal time series, the thorough quantitative analyses, the system (as a perennial grass), the insights about the soil as a reservoir of leaf taxa, the general recent interest in plant-microbiomes to benefit crops and the potential of biofuels as sustainable energy alternatives, that this work would be of interest to the broad readership of Nature Communications. We hope that, given the new modeling results as requested, you will agree!

R3.2

2. Non-stochastic assembly processes. The authors claim that the seasonal dynamics and the accumulation of taxa over time are suggestive of non-stochastic assembly processes (e.g. in lines 98-104). To sustain this statement the authors need to perform simulations of stochastic assembly and compare their results with the results emerging from the stochastic null-models. There are different null models that the authors can use, with different constraints depending on the question asked. For the non-random species richness accumulation, the authors can use different techniques to detrend the data and to test how this alters the observed pattern. For the seasonal data, the authors can consider a common species pool with the empirical numerical abundance of each prokaryote and then subsample from this pool. In general, the authors tend to ascribe differences to host selection. For example, in lines 197-199 the authors claim that differences in relative abundances “of the same taxa across plant hosts, suggesting microbiome selectivity for or by the host plant”. That’s only a possibility. But a more parsimonious explanation that the authors should check for by using null models is that small initial differences between the microbes colonizing each host plant, followed by strong priority effects (see e.g. Sprockett et al. 2018 Nature Rev. Gastro. And Hepat. 15:197, for the human gut microbiota), can explain observed differences without any selectivity by the plant host.

>>>Thank you for this thoughtful comment. We have modeled stochastic assembly from the soil and compared these results with the observed assembly as you suggested (Figure 3). We found that the leaf assembly is deterministic and very different from the random expectation, and furthermore that it differentiates to deterministic quite early in the growing season.

>>>We appreciate the reference to Sprockett et al. 2018 – thank you. However, we believe we do not have data to determine if priority effects are important in this system. Here is the rationale, and we’re happy to have your feedback: First, our study design involved, for each crop, four replicate fields in a random block design. Each field contained two subplots (nitrogen free and fertilized), totaling 8 possible samples per time point. Multiple leaves from several plants were sampled in each field and combined into a representative sample. We were not tracking the assembly on individual plants but rather average assembly for plants in a field that were close in their phenology. So, small differences on individual plants’ communities would not be knowable by our study design. Small differences across individual fields would be knowable, but this is not aligned with the idea of priority effects leading to discrete eventual differences in the individual plant community. We do not expect priority effects to manifest at a field level of observation.

>>>Regardless, the early season samples (May and June) across replicate fields and across crops are statistically similar (Figure 2, Figure S2, PERMANOVA Table 1, Table S3, Figure S4), strongly suggesting that the crops begin with exposure to a highly similar species pool, largely of taxa shared with the soil. Then, within a crop, replicate fields maintain a consistent seasonal trajectory. If priority effects were to manifest at the field level, we may expect at least a few disparate trajectories across crops. Instead, we observe a distinction by crops maximized mid-season and then converged again at senescence. This could be attributable to host selection or general environmental filtering, but it is statistically unlikely that it would be priority effects that happen across crop lines, given 8 replicates per crop, and manifesting at the field level.

R3.3

3. Soils as the most substantial reservoir of leaf microorganisms. That’s a really novel aspect that challenges

prevailing wisdom. However, I am not convinced by the proof provided. The authors do not present data from aerial microbes. The fact that leaf microbes are also present in the soil, but at different abundances, could also be interpreted as if wind transports microbes that then do grow in abundance better in the leaf or in the soil. Hence, stochastic “seedling” from the atmosphere microbes, followed by species sorting (or habitat filtering), could lead to the observed pattern. Again, the way to test for their hypothetical “soil reservoir” hypothesis is to run the null models described above. Even better, the authors could model their hypothesis, using a source-sink metacommunity model, where the soil is the source, and leaves are the sink, plus some simple logistic population growth (or Gompertz model). They could then compare the results of this model with the outcome of an additional stochastic model that tests the “atmosphere reservoir” hypothesis. Then, they can answer the question: which of these two contrasting models better fit the data?

>>>Thank you for this comment, which is similar to comments made by reviewers 1 and 2. Please especially see our responses to R1.4 and R1.5, in which we detail our new analyses including the one you have suggested here. Thank you especially for the suggestion to develop source-sink models to provide more quantitative insights into the observed dynamics, which we have done (Figure 3); these results support our initial hypothesis. We agree that the microbes that were detected abundantly on leaves but in the rare biosphere of soils could have been transported via the air or other vectors like insects, etc, and so we have added text clarifying that we cannot know directionality or mechanism of these taxa’s arrival to the leaf, we only know that almost all of them are detected, typically in the rare biosphere, of associated soil. We have replaced the previous Figure 3 with the new Figure 3 to show the relationship between abundance and detection in the phyllosphere versus the soil. Please also note that we have tempered statements throughout the revised to move away from “origins” to the more neutral term of “reservoir” in the soil, and also have added text to clarify that we cannot know by which the soil-associated taxa were dispersed to the leaf/between the leaf and the soil. All of this is detailed in the revised section starting **L115** with new additions beginning to be detailed **~L133**, and the new source-sink analyses presented starting **L152**.

R3.4

4. Core phyllosphere taxa definition. Core taxa are defined by a threshold for persistence and abundance (lines 151-156). Both thresholds seem arbitrary. Some authors have shown that the criteria for core definition require a systematic exploration (e.g., Astudillo et al. 2017, *Env. Microb.* 19: 1450). Different inclusion criteria (thresholds) can result in different patterns. The question then is: how robust are the patterns displayed by the core, the core microbial networks, and the drivers of phyllosphere assembly, to different inclusion criteria?

>>>Thank you for this comment, and for the reference, which we have added (**L203**). We enthusiastically agree that, though very common in the literature, arbitrary thresholds applied to define core taxa can result in different cores and that an ecological perspective may be more useful. We also agree completely that systematic exploration of the core is important, and we have published on statistical methods to explore and define core taxa (Shade and Handelsman 2012). In this revision and given your encouragement in this direction, we have re-considered how to define a core that leverages the replicated time series information. Thus, we identified a core by including taxa that exhibit 100% occupancy in at least one sampling time point to prioritize those taxa that potentially exhibit dependency on the plant development or seasonal conditions (as seen in all plots at the same time point). Using this approach, we have modestly extended the list of prioritized taxa (Dataset 1), though we note that all previously defined core taxa are still included. As before, we found that these core taxa are persistent over time and highly abundant.

>>>>Please see the explanation of the core inclusion criteria starting **L190**, the results starting **L192** and methods starting **L379**. We also suggest that this method, which leverages time series replication, will be of interest generally to microbiome researchers especially as more microbiome time series are being collected and analyzed.

>>> Please also see our detailed response to R2.3 related to this comment. We confirm that we have systematically explored different inclusion criteria and data series characteristics (e.g., subsampled to different depths, see Table S1, Table S2) and repeatedly find the majority of the same core taxa and all of the same core Classes by abundance and occupancy.

>>> We also present our results rarefied to three different levels (141, 1000, 5000, and 10000 reads per sample, Table S1, Table S2, see also response to R2.17 and R1.19), which will include more observation of the rare biosphere and show that this does not fundamentally alter the patterns exhibited by the core taxa because they are, by definition, not within the rare biosphere. Finally, we have added clarifying text suggesting prioritization of next studies towards these core taxa, and with the exception that there could be others not identified here that also may have importance (**L194**).

Minor

R3.5

- Line 88, and line 306: why this rarefaction cut-off? I assume it is linked to the minimum number or reads per sample in the database. This needs to be confirmed- as the authors say, it is a surprisingly low cut-off, although the relatively low richness of the system can explain this. The authors claim that the results of community structure are consistent between rarefactions to 146 and 500 reads. What aspects of community structure? If so, why not focusing on the rarefactions to 500 reads, providing a larger coverage?

>>>The original rarefaction cut-off was linked to the minimum number of reads in a sample, but we also thought it was appropriate because of the low richness in the leaf communities (fewer than 150 taxa observed at any time point), which did not require a substantial sequencing depth to capture overarching patterns in beta diversity. However, at the requests of reviewers 1 and 2 we have increased to a rarefaction depth of 1,000 reads per sample. This is at the expense of “losing” time points in our data series, and so we also provide the analysis of the full time series at the more limited sampling depth (Figure S2). We also show that the seasonal patterns in beta diversity are unaffected by sampling depth (Figure S2, Table S1 and Table S2).

R3.6

- Line 216: Please provide references to the variance partitioning method used.

>>>At the request of Reviewer 1 (R1.2) to focus the manuscript on the most compelling results, we have omitted the variance partitioning analysis, which was redundant with the PERMANOVA tests (for time, crop/host, Table 1) and EnvFit analysis for abiotic conditions (Table S3).

R3.7

- Line 220-221: It is not clear whether the authors tested explicitly for spatial autocorrelation. They only mention “spatial distance between the plots had no explanatory value”. To me, that only means that they have no distance-decay patterns, but not that spatial autocorrelation is affecting their results within this section.

>>> Thank you. We have added clarifying text to describe our text: “as assessed by distance-decay of beta diversity using a Mantel test with a spatial distance” (**L252**). Borcard and Legendre (2012, *Ecology*) show that the Mantel test is comparable in power and success to Moran’s I (and other methods) in detecting spatial correlation, and is a viable choice of method for such analyses.

>>>Thank you again for your thoughtful and thorough review of the work! Your suggestions have strengthened the piece and we are grateful to have had such positive and constructive reviewers for our work!

References

- Anderson MJ, Ellingsen KE, McArdle BH. Multivariate dispersion as a measure of beta diversity. *Ecology Letters*. 2006 Jun;9(6):683-93.
- Burns AR, Stephens WZ, Stagaman K, Wong S, Rawls JF, Guillemin K, Bohannan BJ. Contribution of neutral processes to the assembly of gut microbial communities in the zebrafish over host development. *The ISME Journal*. 2016 Mar;10(3):655.
- Borcard D, Legendre P. Is the Mantel correlogram powerful enough to be useful in ecological analysis? A simulation study. *Ecology*. 2012 Jun 1;93(6):1473-81.
- Gaston KJ, Blackburn TM, Greenwood JJ, Gregory RD, Quinn RM, Lawton JH. Abundance–occupancy relationships. *Journal of Applied Ecology*. 2000 Sep 1;37:39-59.
- Gobet A, Quince C, Ramette A. Multivariate Cutoff Level Analysis (MultiCoLA) of large community data sets. *Nucleic acids research*. 2010 Jun 14;38(15):e155-.
- Lee SH, Sorensen JW, Grady KL, Tobin TC, Shade A. Divergent extremes but convergent recovery of bacterial and archaeal soil communities to an ongoing subterranean coal mine fire. *The ISME Journal*. 2017 Jun;11(6):1447.
- Oksanen J, Blanchet FG, Kindt R, Legendre P, Minchin PR, O’hara RB, Simpson GL, Solymos P, Stevens MH, Wagner H, Oksanen MJ. Package ‘vegan’. *Community ecology package, version*. 2015 Aug;2(9).
- Ruan Q, Dutta D, Schwalbach MS, Steele JA, Fuhrman JA, Sun F. Local similarity analysis reveals unique associations among marine bacterioplankton species and environmental factors. *Bioinformatics*. 2006 Jul 31;22(20):2532-8.
- R Core Team (2018). R: A language and environment for statistical computing. R Foundation for Statistical Computing, Vienna, Austria. <https://www.R-project.org/>
- Shade A, Handelsman J. Beyond the Venn diagram: the hunt for a core microbiome. *Environmental Microbiology*. 2012 Jan;14(1):4-12.
- Shade A, Jones SE, Caporaso JG, Handelsman J, Knight R, Fierer N, Gilbert JA. Conditionally rare taxa disproportionately contribute to temporal changes in microbial diversity. *MBio*. 2014 Aug 29;5(4):e01371-14.
- Shade A, Dunn RR, Blowes SA, Keil P, Bohannan BJ, Herrmann M, Küsel K, Lennon JT, Sanders NJ, Storch D, Chase J. Macroecology to unite all life, large and small. *Trends in ecology & evolution*. 2018 Sep 9.
- Sloan WT, Woodcock S, Lunn M, Head IM, Curtis TP. Modeling taxa-abundance distributions in microbial communities using environmental sequence data. *Microbial ecology*. 2007 Apr 1;53(3):443-55.
- Vorholt JA. Microbial life in the phyllosphere. *Nature Reviews Microbiology*. 2012 Dec;10(12):828.
- Xia LC, Steele JA, Cram JA, Cardon ZG, Simmons SL, Vallino JJ, Fuhrman JA, Sun F. Extended local similarity analysis (eLSA) of microbial community and other time series data with replicates. *BMC systems biology* 2011 Dec (Vol. 5, No. 2, p. S15). *BioMed Central*.
- Zaneveld JR, McMinds R, Thurber RV. Stress and stability: applying the Anna Karenina principle to animal microbiomes. *Nature microbiology*. 2017 Sep;2(9):17121.

Reviewers' comments:

Reviewer #1 (Remarks to the Author):

I am satisfied that the most important revisions were made by the authors in response to our first set of edits.

Reviewer #2 (Remarks to the Author):

The authors provide a new version of the manuscript with clean and clear introduction and objectives. In addition, they made extensive and appropriate corrections as advised by the three referees. Overall, their manuscript provides novel and pertinent knowledge on the phyllosphere dynamics of two biofuel crop species.

To perfect their work, I have three last comments:

Figure 2A could be improved by modifying the arrow labels location in the figure in another program such as ppt or photoshop.

Figure 4 could be improved by reducing the width of the lines.

Table 1. It is still not clear for the readers which models were tested. Are the models showed in Table 1. single-variable models or were the models tested with all variables together?

Reviewer #3 (Remarks to the Author):

2nd review of Grady et al. "Assembly and seasonality of core phyllosphere microbiota on perennial biofuel crops" for Nature Communications.

The authors have done a thorough revision of the manuscript. I believe they have thought to a great extent about my concerns (and those of the other 2 reviewers), and provided much more convincing arguments for their interpretations, even if I do not agree fully with some of their interpretations. Overall, the manuscript is much clearer, the arguments better supported, and its relevance and originality better stated. Below I detail my views on the improvements made in the manuscript to deal with my initial concerns.

- Novelty and conceptual advance: The Introduction and motivation of the paper are much clearer and focused now. From a restricted host-microbiome perspective, I still believe the paper is not providing any major conceptual advance. Sure the dataset is very good in terms of temporal and spatial resolution. And the (partial) demonstration that each host is "selecting" a different set of microbes with respect to the null expectation is a great result. But the novelty and originality of this paper is the combination of basic, fundamental host-microbiome analyses, with applied potential of biofuels.

- Non-stochastic assembly processes and soil as main reservoir: The authors have done a very good work using one of the possible source-sink models, and the additional Figure 3 provided helps a lot. The authors convincingly demonstrate the deterministic assembly from the soil reservoir. However, I still have the same concern regarding the potential relevance of priority effects. I still believe the authors can provide evidence for or against priority effects, even at the field level (given they pooled leaves from different plant individuals). Testing for priority effects does not require individual-level microbiomes, and I believe it is a key piece of this paper. I do not see a strong selection effect at the host species level- amongst the explanatory variables, host

species has the lowest explanatory power (as seen in Table 1). Time is key- stage in plant development. But, still, priority effects can explain the dynamics. I do not want to be stubborn with this argument, but it is key for their interpretation and possible application. I still believe the authors should check for the presence of priority effects to determine whether what happens at the very beginning of the growing season (most likely coming from the soil reservoir) determines the assembly dynamics over time, or whether it is the continuous seeding from the source soil community through time what determines microbiome dynamics.

- Core taxa definition: I like very much your definition of core taxa. It is conservative and robust.

- I have an additional major comment. Why the authors use OTU clustering (97%) and not ASVs, as recently recommended by the microbiome research community (e.g. Knight et al. 2018 "Best practices for analysing microbiomes" Nature Reviews Microbiology <https://doi.org/10.1038/s41579-018-0029-9>)? I don't see the need to use OTUs- specially in such a low diversity system.

Some other minor comments

Line 48: Resilience- its use here is vague- what do the authors mean exactly?

Line 253: please rewrite

Line 265: Please include the reference in the main text

Figure 4: The authors should think on reducing the number of panels here- I found it overcomplicated.

>>>Thank you to the reviewers for their positive comments on the revised manuscript! We are pleased to report that we have addressed all minor points and added additional discussion with reference to analysis and figures to address the major point of priority effects. Each reviewer query has been numbered and these numbers are used to link similar reviewer comments and author responses. Author responses are indented and indicated also with >>>. Line numbers refer to the numbers in the tracked changes (marked up) document.

Reviewers' comments:

Reviewer #1 (Remarks to the Author):

I am satisfied that the most important revisions were made by the authors in response to our first set of edits.

>>>Thank you so much for the positive response!

Reviewer #2 (Remarks to the Author):

The authors provide a new version of the manuscript with clean and clear introduction and objectives. In addition, they made extensive and appropriate corrections as advised by the three referees. Overall, their manuscript provides novel and pertinent knowledge on the phyllosphere dynamics of two biofuel crop species.

>>>Thank you so much for the positive response!

To perfect their work, I have three last comments:

Figure 2A could be improved by modifying the arrow labels location in the figure in another program such as ppt or photoshop.

>>>We have improved the arrow labels in Figure 2A.

Figure 4 could be improved by reducing the width of the lines.

>>>We have reduced the width of the lines in Figure 4

Table 1. It is still not clear for the readers which models were tested. Are the models showed in Table 1. single-variable models or were the models tested with all variables together?

>>>We have added text to the Table legend to clarify the models tested.

Reviewer #3 (Remarks to the Author):

2nd review of Grady et al. "Assembly and seasonality of core phyllosphere microbiota on perennial biofuel crops" for Nature Communications.

The authors have done a thorough revision of the manuscript. I believe they have thought to a great extent about my concerns (and those of the other 2 reviewers), and provided much more convincing arguments for their interpretations, even if I do not agree fully with some of their interpretations. Overall, the manuscript is much clearer, the arguments better supported, and its relevance and originality better stated. Below I detail my views on the improvements made in the manuscript to deal with my initial concerns.

>>>Thank you for the thoughtful comments and we're glad that you believe the manuscript is improved.

- Novelty and conceptual advance: The Introduction and motivation of the paper are much clearer and focused now. From a restricted host-microbiome perspective, I still believe the paper is not providing any major conceptual advance. Sure the dataset is very good in terms of temporal and spatial resolution. And the (partial) demonstration that each host is "selecting" a different set of microbes with respect to the null expectation is a great result. But the

novelty and originality of this paper is the combination of basic, fundamental host-microbiome analyses, with applied potential of biofuels.

>>>Thank you for this comment. We strive to make the novelty and originality clear in the introduction text.

- Non-stochastic assembly processes and soil as main reservoir: The authors have done a very good work using one of the possible source-sink models, and the additional Figure 3 provided helps a lot. The authors convincingly demonstrate the deterministic assembly from the soil reservoir. However, I still have the same concern regarding the potential relevance of priority effects. I still believe the authors can provide evidence for or against priority effects, even at the field level (given they pooled leaves from different plant individuals). Testing for priority effects does not require individual-level microbiomes, and I believe it is a key piece of this paper. I do not see a strong selection effect at the host species level- amongst the explanatory variables, host species has the lowest explanatory power (as seen in Table 1). Time is key- stage in plant development. But, still, priority effects can explain the dynamics. I do not want to be stubborn with this argument, but it is key for their interpretation and possible application. I still believe the authors should check for the presence of priority effects to determine whether what happens at the very beginning of the growing season (most likely coming from the soil reservoir) determines the assembly dynamics over time, or whether it is the continuous seeding from the source soil community through time what determines microbiome dynamics.

>>> We're glad that the source-sink models are helpful! Thank you! We think this has really strengthened the work – thank you for suggesting it.

>>>Thank you also for your push for us to include discussion of priority effects. We have considered the mentioned paper in the previous review (Sprockett et al. 2018) and we could not identify an appropriate example therein that the reviewer desires us to emulate. We very respectfully offer a difference of opinion as far as whether it may be appropriate to consider priority effects at the field level, because we question whether it is ecologically informative to think about priority effects aggregated across multiple individual hosts. But, even at with field aggregation, we found no evidence for strong priority effects in this study. Our reasoning is as follows:

>>>Priority effects occur when differences in colonization history result in different community assembly outcomes. While we cannot claim that colonization histories had no effect on phyllosphere successional patterns, the data shown in Fig. S3 and Fig. S5 suggest that any such effects were relatively minor. Specifically, beta-dispersion among plants of the same species decreased over time (Fig. S3), and Bray-Curtis dissimilarity between switchgrass and miscanthus cohorts decreased over time (Fig. S5 [*but see note below]). Both lines of evidence point to phyllosphere community *convergence* over time, despite early some differences in community composition, suggesting that colonization history per se had little effect on successional trajectories.

>>>Importantly, this is not to say that early colonists did not modify the phyllosphere environment such that particular species were subsequently favored or disfavored, but that early colonists did not modify the phyllosphere environment differentially and/or to a meaningful degree. This is supported by the data in Fig. 4. The majority of the early colonizers were rare or absent by the end of the growing season (Fig. 4G-I), and so they did not modify the phyllosphere environment (i.e., exert priority effects) to ensure and/or enhance their competitive dominance (i.e., niche construction, or niche preemption *sensu* Fukami 2015). We have added this text to the paper in the discussion section and hope that this detailed and clear discussion of the logic is satisfactory (Discussion L390, two paragraphs immediately prior to conclusions).

- Core taxa definition: I like very much your definition of core taxa. It is conservative and robust.

>>>Thank you for the positive feedback! We think this will help others in the field to define core microbiome as well.

- I have an additional major comment. Why the authors use OTU clustering (97%) and not ASVs, as recently

recommended by the microbiome research community (e.g. Knight et al. 2018 “Best practices for analysing microbiomes” Nature Reviews Microbiology <https://doi.org/10.1038/s41579-018-0029-9>)? I don't see the need to use OTUs- specially in such a low diversity system.

>>>Thank you for this comment. Indeed, we have thought a lot about OTU definitions, for example in observing the strong temporal pattern and also in defining core microbiome. Other work has shown that overarching patterns in beta-diversity can be consistent regardless of the use of 100% OTUs or 97% (Glassman and Martiny 2018). Therefore, we tested overarching temporal patterns in beta diversity for our own dataset using both 100% and 97% OTUs and found them to be statistically indistinguishable with both PROTEST and Mantel (Correlation in a symmetric Procrustes rotation 0.9075, p-value 0.001, 999 permutations Mantel correlation 0.6966, p-value 0.001, 999 permutations.). This is an important outcome because of the longitudinal design of the study and importance of time for explaining community shifts. So, the use of either OTU definition would not impact the observation of these overarching temporal patterns or interpretation of these patterns.

>>>Our second focus was to identify core taxa over time. We reason that any undetected sequencing errors and <100% sequence identity among multiple copies of the 16S rRNA gene from the same genome are clustered with the parent sequence when using 97% definition. We were concerned that this would inflate the membership of the core and add redundancy. Therefore, we decided that it would be most prudent to proceed conservatively with the 97% OTUs to define the core. Additionally, we provide the representative sequence of each core OTU in Supporting Dataset 1 so that any researcher interested in 100% OTUs can use this information with the raw dataset (on IMG) to split the 97% clusters.

>>>We agree that this is an interesting and ongoing issue of using arbitrary taxonomic units. In a biological sense, it is likely that there is a not a “one size fits all” OTU definition for all lineages (e.g., 97% would be most appropriate for some and 100% would be appropriate for others). But, providing all of the data publicly will allow others to reproduce the analysis while varying the %identity.

Some other minor comments

Line 48: Resilience- its use here is vague- what do the authors mean exactly?

>>>Thank you for this comment. We have clarified by adding “resilience to environmental stress”.

Line 253: please rewrite

>>>We have re-written it.

Line 265: Please include the reference in the main text

>>>How we should act on this comment is unclear to us because this line is indeed in the main text, and there is a reference in (previous tracked changes version) Line 264 but not Line 265.

Figure 4: The authors should think on reducing the number of panels here- I found it overcomplicated.

>>>Thank you for this suggestion. We have decided to keep the number of panels, but have improved the line widths as suggested by Reviewer 2.

REVIEWERS' COMMENTS:

Reviewer #3 (Remarks to the Author):

3rd review of of Grady et al. "Assembly and seasonality of core phyllosphere microbiota on perennial biofuel crops" for Nature Communications.

I had 2 remaining concerns with this ms in my 2nd review.

Lack of priority effects. I have a different opinion than that of the authors in regards to whether one can perform a priority effects analysis at the field level. One could aggregate individual hosts and compare rank-abundance distributions through time. However, I am happy to disagree here, and if the authors want to focus their priority effects discussion on convergence over time at the host level, it's their decision. I agree the authors have powerful arguments here, yet I believe some readers will find the discussion uncomplete. My suggestion would be for the authors to think about this again, but I am happy with any decision they make here.

Clustering at the 97 or 100 percent. I would suggest the authors are very clear about their reasoning on their selection in the paper. First, be clear in the main text about the robustness of beta-diversity pattern to the cut-off used. Second, be explicit about your conservative approach to select the cut-off for the core definition.

REVIEWERS' COMMENTS:

Reviewer #3 (Remarks to the Author):

3rd review of of Grady et al. "Assembly and seasonality of core phyllosphere microbiota on perennial biofuel crops" for Nature Communications.

I had 2 remaining concerns with this ms in my 2nd review.

Lack of priority effects. I have a different opinion than that of the authors in regards to whether one can perform a priority effects analysis at the field level. One could aggregate individual hosts and compare rank-abundance distributions through time. However, I am happy to disagree here, and if the authors want to focus their priority effects discussion on convergence over time at the host level, it's their decision. I agree the authors have powerful arguments here, yet I believe some readers will find the discussion incomplete. My suggestion would be for the authors to think about this again, but I am happy with any decision they make here.

>>Thank you again for this comment, and we understand that this is a topic of great interest to the reviewer; we also agree that priority effects can be generally important for community assembly and should be considered. There is a clarifying point that we would like to make, which is that we have already aggregated across individual hosts for our analyses (this was part of the study design - see Methods), and it is impossible to segregate hosts because they were sampled as a pool of leaves representative of the field. So what we have reported is what we feel is the best that can be done, aggregating across individuals to the field level. The next step would be to aggregate across fields, which, again, we feel is not appropriate conceptually for understanding priority effects. We hope that the discussion that we added will suffice.

Clustering at the 97 or 100 percent. I would suggest the authors are very clear about their reasoning on their selection in the paper. First, be clear in the main text about the robustness of beta-diversity pattern to the cut-off used. Second, be explicit about your conservative approach to select the cut-off for the core definition.

>>>Thank you, we have added the suggested text starting line 387 and also at line 436 to explicitly clarify that we performed this analysis and to provide our reasoning for our core definition.